# Single-cell and spatial transcriptomics reveal post-translational modifications in osteosarcoma progression and tumor microenvironment

**Yue Cui, Yue Wu, Dongyu Jiang, Tao Ding** *

The Affiliated Wuxi People's Hospital of Nanjing Medical University, Wuxi, Jiangsu Province, China

\* dingtao@njmu.edu.cn

## Abstract

Emerging evidence suggests that post-translational modifications (PTMs) contribute to osteosarcoma pathogenesis, yet their exact molecular roles require further elucidation. Using the AddModuleScore method, we classified tumor cells on the basis of PTMs scores via single-cell RNA sequencing (scRNA-seq). The robust cell type decomposition (RCTD) approach was then applied to map these single-cell groupings onto spatial transcriptomics (ST), enabling the analysis of cell dependencies and the identification of distinct tumor cell subtypes. A prognostic model was constructed using bulk transcriptomic data to predict therapeutic outcomes in immunotherapy. Laboratory experiments were carried out to confirm the biological function of vimentin *(VIM)*. PTMs scores were significantly elevated in tumor cells, stratifying osteoblastic cells (os) into two clusters: PTMs highos and PTMs lowos. The PTMs highos phenotype exhibited pronounced malignant characteristics and closely interacted with fibroblasts in both scRNA-seq and ST analyses. A set of ten hub genes was identified, forming a consensus machine learning-derived post-translational modification gene signature (CMDPTMS) with strong prognostic predictive capability. Although the high-CMDPTMS group (over the median risk score) was linked to poor outcomes and diminished benefit from immunotherapy, seven drugs were identified that may offer therapeutic promise for these patients. Finally, we confirmed that *VIM* can inhibit the growth and migration of OS cells. In summary, by integrating bulk RNA-seq, scRNA-seq, and ST, we introduced that CMDPTMS may serve as a powerful tool for enhancing OS prognosis prediction and optimizing immunotherapy strategies.

## Introduction

Osteosarcoma (OS) is an exceptionally aggressive primary bone malignancy primarily observed in children and young adults; OS represents approximately 5% of all pediatric malignancies and accounts for a significant proportion (20%) of bone-related cancers [1]. Despite the availability of chemotherapy, radiotherapy, and

---

**Data availability statement:** The datasets used in this study are publicly available from the TCGA database (http://cancergenome.nih.gov/) and the GEO database (https://www.ncbi.nlm.nih.gov/geo/). All relevant data are provided within the manuscript and its Supporting information files.

**Funding:** The author(s) received no specific funding for this work.

**Competing interests:** The authors have declared that no competing interests exist.

surgical intervention as the most effective clinical treatment modalities, OS continues to demonstrate a poor clinical prognosis, primarily because of its aggressive biological behavior and high metastatic potential [2]. Recent advances in OS research have revealed multiple promising therapeutic targets, including receptor tyrosine kinases, immune checkpoint inhibitors, and key metabolic pathways, which may revolutionize the treatment paradigm for this aggressive malignancy [3,4]. Advances in genomics and transcriptomics have led to the identification of multiple molecular subtypes of OS, which may display distinct treatment responses and prognostic outcomes [5]. Understanding the molecular characteristics of OS can provide opportunities to refine personalized treatment strategies and improve patient clinical results.

Post-translational modifications (PTMs) represent crucial regulatory mechanisms that modulate protein properties and functions following biosynthesis [6]. An increasing body of evidence has demonstrated that diverse PTMs, including acetylation, methylation, small ubiquitin-like modifier modification (SUMOylation), and ubiquitination, play pivotal roles in the pathogenesis of OS. Notably, H3K27 acetylation-mediated activation of COL6A1 has been shown to promote pulmonary metastasis in OS through dual regulatory mechanisms involving STAT1 suppression and subsequent activation of cancer-associated fibroblasts within the lung microenvironment [7]. Furthermore, ALKBH1-regulated N6-methyladenosine modification of TRAF1 mRNA significantly enhances OS progression by simultaneously promoting tumor proliferation and metastatic potential [8]. Moreover, the knockdown of SUMO-specific protease 1 markedly inhibits cell viability, accelerates apoptosis, and suppresses invasive capacity by influencing epithelial-mesenchymal transition in OS cells [9]. Additionally, RFWD2, which functions as an E3 ubiquitin ligase, has been identified as an oncogenic factor that regulates OS cell proliferation and modulates sensitivity to cisplatin treatment [10]. Collectively, these results underscore the pivotal role of PTMs in OS pathogenesis and highlight their potential as promising therapeutic targets for OS intervention [11,12].

In this study, we employed an integrative multi-omics approach, integrating single-cell RNA sequencing (scRNA-seq), spatial transcriptomics (ST), and consensus machine learning algorithms, to systematically investigate the multifaceted roles of PTMs in OS. Our comprehensive analysis focused on elucidating the influence of PTMs on drug response, tumor immune microenvironment, immunotherapy efficacy, and clinical outcomes. By leveraging these advanced technologies and analytical frameworks, we aimed to elucidate the underlying molecular mechanisms of PTMs in OS pathogenesis and progression. The findings from this integrated approach provide novel insights into molecular stratification, prognostic prediction, and the formulation of tailored therapeutic approaches for OS patients.

## Methods

### Acquisition of PTMs gene set

Using resources from extensive literature analysis, the Kyoto Encyclopedia of Genes and Genomes (KEGG) database (https://www.kegg.jp/), and Gene Set Enrichment Analysis (GSEA) platform, we delineated 11 distinct types of PTMs. These

encompass ubiquitination, methylation, phosphorylation, glycosylation, acetylation, SUMOylation, citrullination, neddyla- tion, palmitoylation, ADP-ribosylation, and succinylation. Gene assemblies corresponding to each PTMs variant were systematically compiled and cataloged in S1 Table in S1 File, encompassing a total of 3401 genes.

### Single-cell RNA sequencing (scRNA-seq) data acquisition and processing

The scRNA-seq datasets GSE152048, GSE200529, GSE198896, GSE270231, GSE162454, GSE250015, GSE234187, GSE169396, and GSE217792 were accessed from the GEO repository. Seurat objects were generated using the "Seurat" R package, which integrates only high-quality cells for subsequent analyses [13,14]. All data were log-normalized and batch-corrected using the harmony algorithm, after which 1,500 genes with the highest coefficient of variation were filtered out [15]. The reduction in dimensionality was accomplished through the application of the uniform manifold approximation and projection (UMAP) method [16,17]. Marker genes for cell clusters were identified by the FindAllMarkers function, and cell-type annotation was performed according to previously published article [18]. Furthermore, the "AddModuleScore" function in Seurat was used to quantify PTMs activity in each cell [19].

### Spatial transcriptomics (ST) analysis

ST data were sourced from a previous study [20] and processed using the Seurat R package. Data normalization was per- formed via log transformation. Clustering of ST points was performed using the FindClusters and FindNeighbors functions. Clusters were initially distinguished by hematoxylin-eosin (H&E) staining and further annotated using unsupervised clus- tering analysis. To investigate cancer cell progression, the Monocle2 package was employed to order cancer cell popula- tions along their developmental trajectories in pseudotime [21,22].

### Spatial interaction analysis

The mistyR package (v1.6.1) was employed to evaluate spatial interactions within the integrated ST-scRNA-seq dataset, including single spots and adjacent regions within radii of 5 and 15 spots [23].

### Bulk data collection

OS tissue gene expression data were retrieved from TARGET-OS and GEO databases. These datasets were merged using the "limma" and "sva" packages to form a comprehensive cohort [24]. The TARGET-OS dataset was designated as the training cohort (87 patients), while GSE39055, GSE16091, and GSE21257 were utilized as the validation cohort (88 patients), incorporating clinical information, somatic mutation data, and RNA sequencing data.

### Establishment of a consensus machine learning-derived post-translational modification gene signature (CMDPTMS)

Ninety-seven prognosis-related PTMs were determined by univariate Cox regression analysis. On the basis of the liter- ature, a CMDPTMS with high accuracy and stability was developed using the "ML.Dev.Prog.Sig" function in the Mime package [25], integrating 117 model combinations with K-fold cross-validation in the TARGET-OS cohort. The concor- dance index (C-index) was calculated for each model, and the optimal model was selected by maximizing AUC using the "cal_AUC_ml_res" function. The hub genes of CMDPTMS were identified using the "ML.Corefeature.Prog. Screen" function in the Mime package.

### Prognostic evaluation and clinical application of CMDPTMS

The samples in the training and validation cohorts were scored using CMDPTMS, classifying them into high- and low-CMDPTMS clusters. KM survival curves were generated to assess prognostic significance. To comprehensively

evaluate prognostic indicators of OS, we identified 16 clinically relevant molecular features through a systematic literature review. For each patient sample, a quantitative risk score was computed using established weighting coefficients derived from prior studies. The predictive performance of these molecular signatures was subsequently validated across all study cohorts through C-index analysis. CMDPTMS performance was evaluated using AUC, ROC curves, and KM analysis. Accuracy was assessed via C-index and calibration curves, whereas clinical benefits were estimated using DCA.

### Immunotherapy response and immune-omics analysis based on CMDPTMS

Using the IOBR package, we compiled published immune-related signatures [26]. Enrichment scores were computed for each sample, facilitating a comparative immunological assessment of high- and low-CMDPTMS subgroups. Differences in tumor neoantigen burden (TNB) and tumor mutational burden (TMB) between these subgroups were evaluated. To predict immunotherapy response, we analyzed patient survival data following immunotherapy responses and incorporated tumor immune dysfunction and exclusion (TIDE) algorithm and tumor immunophenotype (TIP) algorithm [27,28]. The gene set enrichment analysis (GSEA) method was used to examine pathways that were significantly enriched in each CMDPTMS via the "clusterProfiler" R package (4.10.1) [29].

### Therapeutic response prediction

Anticancer drug sensitivity in high- and low-CMDPTMS groups was compared using $IC_{50}$ values estimated via oncoPredict package [30]. Drug sensitivity metrics were sourced from CTRP v.2.0 and PRISM Repurposing datasets (19Q4). The AUC of dose-response curve was used as a sensitivity measure.

### Cell culture and transfection

MG-63 and U-2 OS (OS cells) and hFOB 1.19 (normal osteoblastic cells) were obtained from Pricella Biotechnology Co., Ltd. The above cells were cultured in their respective cell-specific culture media (Pricella) and maintained in an incubator at 37°C with 5% $CO_2$. To transiently suppress VIM expression, we delivered siRNA (GenePharma) into the cells using Lipofectamine 3000. The detailed siRNA sequences are provided in S2 Table in S1 File.

### RNA extraction and real-time quantitative polymerase chain reaction

Total RNA was extracted using an RNA extraction kit (Vazyme Biotech Co., Ltd), followed by cDNA synthesis with a cDNA synthesis kit (Vazyme). qPCR was performed using a qPCR kit (Vazyme) on a Roche LightCycler 480II system. The quantification of target gene expression levels was determined through the "delta-delta Ct" analytical approach [31]. The primers used in this study are listed in S3 Table in S1 File.

### CCK-8 assays

In total, $3 \times 10^3$ cells were plated into each well of a 96-well plate and incubated for 72 hours. Cell viability was measured at 24-hour intervals using a CCK-8 kit (Beyotime).

### Transwell assay

To assess the migratory capacity of MG-63 and U-2 OS osteosarcoma cell lines, a Transwell invasion assay was performed. The cells ($1 \times 10^4$) were seeded in the upper compartment of a 24-well Transwell insert, while the lower compartment contained Dulbecco's modified Eagle's medium supplemented with 15% fetal bovine serum as a chemoattractant. Following a 24-hour incubation period, invasive cells were fixed with 4% paraformaldehyde for 20 minutes and subsequently stained with 0.1% crystal violet solution for 30 minutes. After washing with phosphate-buffered saline, the migrated cells were visualized and imaged using an Olympus microscope (Japan).

## Statistical analysis

All statistical evaluations were conducted using R software (v4.3.2). *P*<0.05 were deemed statistically significant.

## Results

### Post-translational modifications characteristic in single-cell transcriptome

For our scRNA-seq analysis, we examined 69,342 cells derived from 37 OS samples and 13 normal bone tissue samples. Initial data processing involved the calculation of cell cycle scores using the Seurat package's CellCycleScoring function to mitigate batch effects (Fig 1A). Subsequent clustering analysis, conducted with a resolution parameter of 1.5, identified 49 distinct cellular clusters (Fig 1B). By using established marker genes, we classified these clusters into seven principal cell populations: myeloid cells, fibroblasts, osteoblastic cells, T cells, NKT cells, endothelial cells, and B cells (Fig 1C,D). To evaluate PTMs activity across these cellular populations, we used the "AddModuleScore" function within the Seurat package. Our findings demonstrated significant enrichment of PTMs specifically in osteoblastic cells within tumor tissues (Fig 1E).

### Pseudotemporal analysis and intercellular communication analysis

To elucidate the dynamic role of PTMs in tumor development, we performed pseudotemporal analysis. This analysis revealed temporal expression patterns of PTMs at different stages of tumor progression, with molecular markers including LTF, PADI4, and VIM exhibiting high expression during the neoplastic phase (Fig 2A). To further delineate the evolutionary trajectory of osteoblastic cell subpopulations, we conducted pseudotemporal and cellular trajectory analyses [32]. We identified two distinct subpopulations (Fig 2B), revealing a transitional pattern from a cluster predominantly present in OS samples (cluster 7, 10, 17, 29, and 30).

On the basis of PTMs expression levels, all osteoblastic cells (os) are categorized into two subgroups: PTMs-high osteoblastic cells (PTMshighos) and PTMs-low osteoblastic cells (PTMslowos). Our analysis revealed that clusters 7 and 17, which were dominant in PTMslowos, transitioned into clusters 6 and 8, which were enriched in PTMshighos (Fig 2C). GSEA revealed that the Notch, Hedgehog, and Wnt pathways-abnormally activated pathways that facilitate tumor progression-were markedly enriched in high-PTMshighos compared with PTMslowos (S1 Fig A, B in S1 File). To elucidate the temporal regulatory mechanisms of PTMs expression profiles at single-cell resolution, we conducted trajectory analysis utilizing the monocle algorithm (Fig 2D–F). Subsequent functional annotation of PTMshighos cells via monocle-based trajectory inference revealed significant enrichment of the adipocytokine signaling within this cellular subset (S2 Fig A in S1 File). Additionally, cell-cell communication analysis via a circle plot (S2 Fig B in S1 File) and ligand-receptor interaction heatmaps (S2 Fig C in S1 File) revealed significant differences between PTMshighos and PTMslowos, including IL6−IL6R, IL6−IL6ST, and TNFSF10−TNFRSF10B, which are predominantly mediated by fibroblasts- key stromal components that play a crucial role in driving primary tumor growth and metastasis. Finally, to elucidate ligand-receptor-transcription factor interactions, we generated Sankey network visualizations (S2 Fig D in S1 File). The above results reveal that PTMshighos is positively correlated with tumor evolution, not only through activation of the Notch, Hedgehog, Wnt, and adipocytokine signaling pathways, but also via enhanced cellular interactions between PTMshighos and fibroblast cells.

### Robust cell type decomposition (RCTD) deconvolution analysis

We applied RCTD deconvolution analysis to quantify PTMs expression levels across various osteoblastic cell populations within OS tissue sections [33]. Following spatial single-cell RNA-seq data integration, all osteoblastic cells were classified into two subgroups on the basis of PTMs expression, and eight distinct cell types were identified (Fig 3A). Cell-type composition analysis highlighted the crucial role of intra and para_15 regions in modulating cellular interactions and spatial organization (Fig 3B). Using the mistyR package, we analyzed intra-, juxta-, and para-regional interactions,

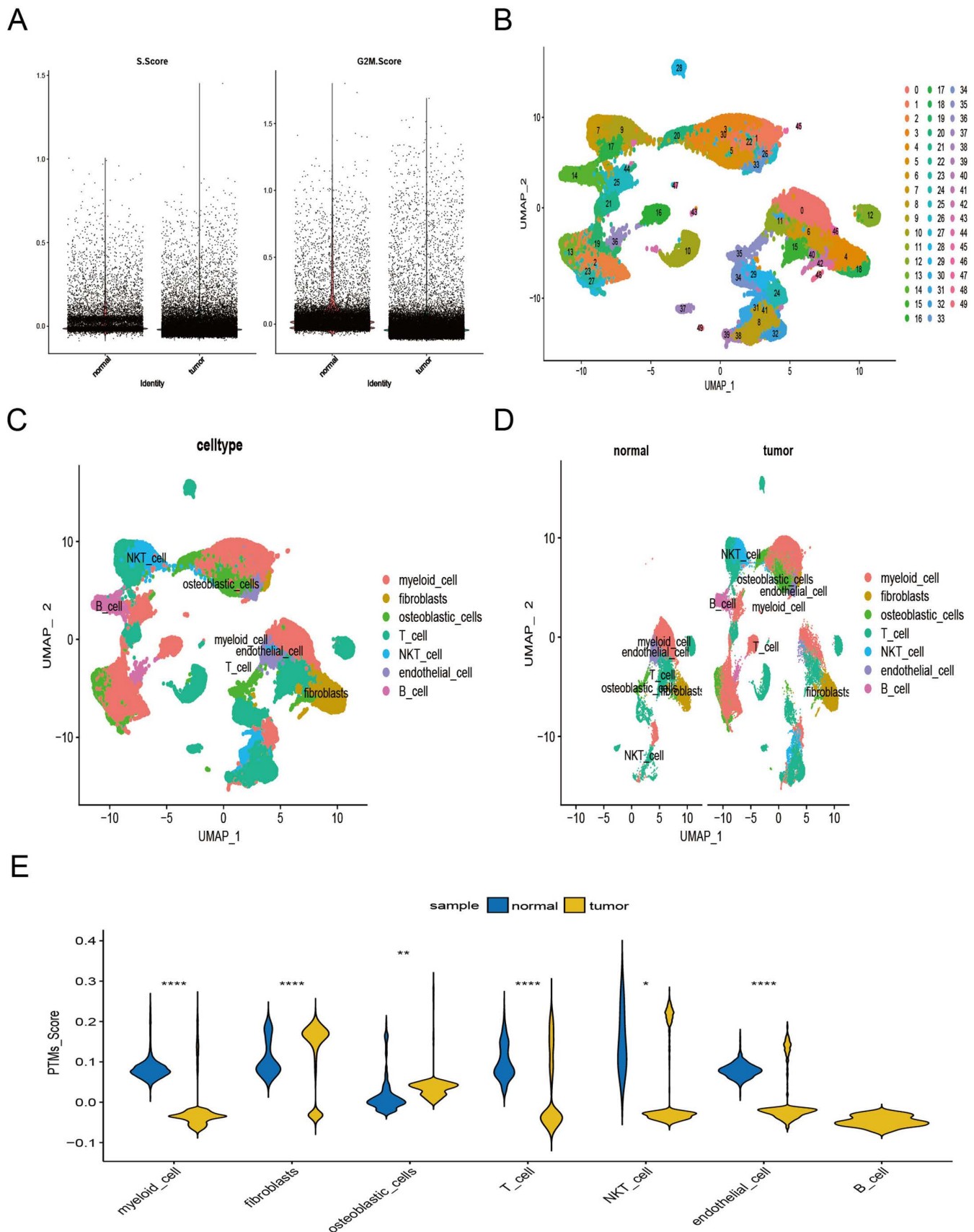

**Fig 1. Characteristics of post-translational modifications (PTMs) in single-cell RNA sequencing (scRNA-seq) of osteosarcoma (OS).** (A) Batch effects were mitigated using the CellCycleScoring function. (B) A UMAP plot visualizes 49 distinct cell subpopulations. (C, D) UMAP plots depict the distributions of 7 major cell types. (C) PTMs activity scores across individual cells in scRNA-seq data. (D) Results of spot clustering analysis in ST data. (E) PTMs activity scores across individual cells in scRNA-seq data.

revealing spatial proximity patterns of PTMshighos and PTMslowos with other cell types. PTMshighos populations were closely associated with fibroblasts and T cells, indicating enhanced intercellular communication with these cells (Fig 3C–H). In contrast, PTMslowos cells were preferentially localized near immune cells, particularly B cells, contributing to an immune-enriched microenvironment. These observations indicate that PTMshighos and PTMslowos exert differential regulatory functions in modulating the tumor microenvironment (TME) in OS.

## Constructing and validating of a CMDPTMS

To quantitatively assess osteoblastic cell abundance using PTMs signatures and enhance prognostic prediction in OS, we developed a CMDPTMS through an unbiased machine-learning framework. We first performed univariate Cox proportional hazards regression to identify 97 PTMs significantly associated with OS prognosis, on the basis of TARGET-OS and GEO-OS datasets. These PTMs were subsequently integrated into an ensemble model constructed from 117 algorithmic combinations, with predictive performance evaluated via the mean C-index (Fig 4A). Among these models, the top five highest-ranked combinations included StepCox[forward] + Ridge, StepCox[forward] + plsRcox, StepCox[forward] + Enet[α = 0.1], StepCox[forward], and RSF + GBM.

Time-dependent ROC curve analysis was employed to validate the predictive performance of the algorithms, revealing that the StepCox[forward] + Ridge model demonstrated superior predictive capability, as evidenced by achieving the highest mean area under the curve (AUC) among the top five algorithms with the highest C-index at both 1-year and 3-year (Fig 4B–D). We termed the StepCox[forward] + Ridge model as CMDPTMS, which incorporates FZD8, IBTK, PPP1CC, VPS25, BAG2, TRIM35, UBE3B, USP51, BRD4, and VIM, demonstrated robust predictive performance, with a mean C-index of 0.74, along with area under the curve values of 0.775 at one year, 0.8 at three years, and 0.81 at five years (Fig 5A–C). ROC analysis further validated CMDPTMS, with 1-, 3-, and 5-year AUCs exceeding 0.7 (Fig 5D). Moreover, high- CMDPTMS patients had poor clinical outcomes in the TARGET-OS and GEO cohorts (Fig 5E,F).

## Prognostic values of CMDPTMS

To facilitate a rigorous benchmarking analysis, we performed an extensive review of recent publications (2020–2025) to identify comparable prognostic signatures in osteosarcoma. Comparative analyses across both the TARGET-OS and GEO-OS cohorts demonstrated that CMDPTMS exhibited superior predictive accuracy, as measured by the C-index, compared with nearly all reference models (S3 Fig A, B in S1 File). To assess the independent prognostic significance of the risk score (RS) generated from CMDPTMS in OS patients, we conducted a comprehensive nomogram. The nomogram exhibited good calibration and predictive accuracy (S4 Fig A in S1 File). The calibration curve confirmed that the nomogram's predictions were consistent with actual outcomes (S4 Fig B in S1 File). DCA further revealed that the nomogram may offer improved prognostic accuracy for OS patients (S4 Fig C-E in S1 File). These findings collectively indicate that CMDPTMS has the potential to function as a significant prognostic indicator for OS.

## Identification of the immune characteristics of CMDPTMS

By employing the "IOBR" R package, we performed an in-depth examination of TME in OS. Our findings revealed that the infiltration levels of immune cells, such as T cell exhaustion, plasma cells, fibroblasts, and MDSC were markedly elevated in high-CMDPTMS patients than in low-CMDPTMS patients (Fig 6A). Immunosuppressive molecular markers and immune

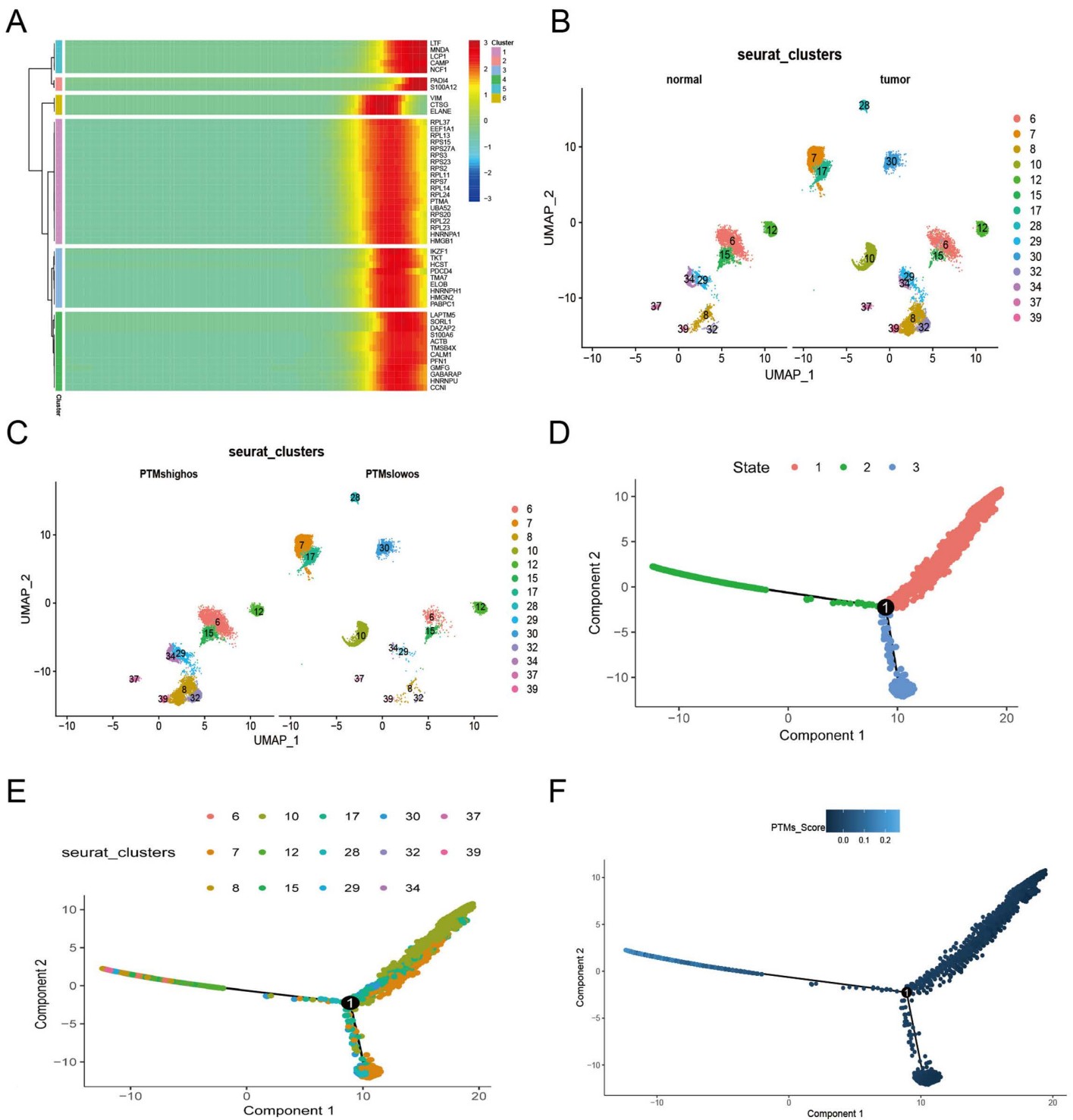

**Fig 2. Pseudotemporal and intercellular communication analyses.** (A) Developmental heatmap of PTMs in osteoblastic cells, illustrating temporal progression (left to right) and expression levels (low: blue; high: red). (B) Classification of osteoblastic cells into normal and OS groups. (C) Classification into PTMs-high (PTMshighos) and PTMs-low (PTMslowos) subgroups. (D, E) Pseudotemporal trajectory analyses of osteoblastic cells, with the developmental starting point at the left root. (F) Branch-specific progression mapping within the PTMs score+ osteoblastic cell subpopulation.

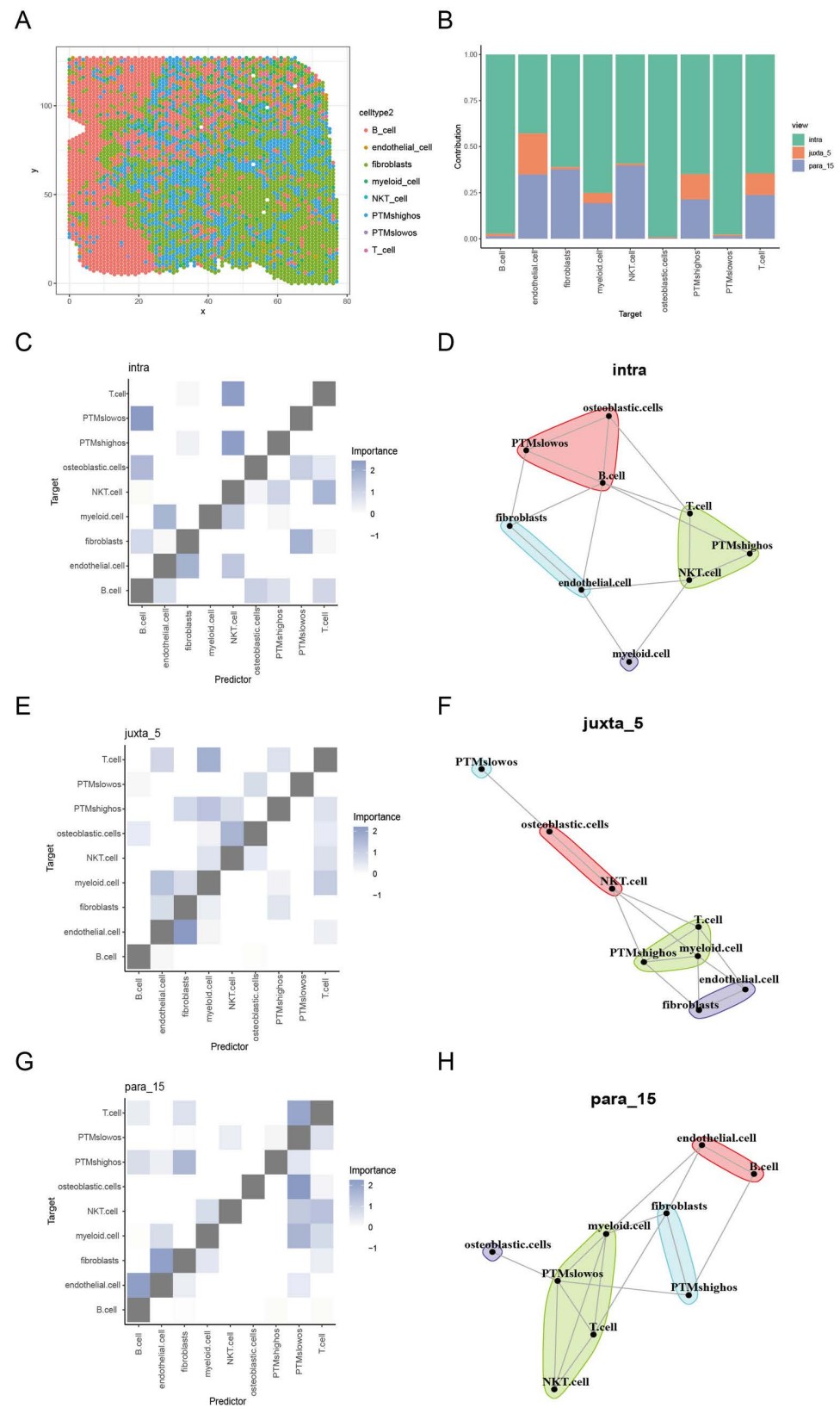

**Fig 3. The ST, spatial gene score, and spatial PTMs map of OS.** (A, B) High-resolution tissue images of OS, highlighting distinct cell clusters identified via scRNA-seq. Clusters are classified based on PTMs scores, dividing osteoblastic cells into PTMshigh and PTMslow groups, with spatial distribution mapped onto tissue images. (C, D) Spatial interactions between PTMshigh/PTMslow osteoblastic cells and other cell types within the core region. (E, F) Spatial interactions within a 5-unit radius. (G, H) Spatial interactions within a 15-unit radius.

checkpoint were enriched predominantly in low-CMDPTMS patients, whereas markers associated with immunoexclusion, such as CAF and TGFb family members, were enriched primarily in the high-CMDPTMS group (Fig 6B,C). These findings indicate that high-CMDPTMS OS is characteristic of a "cold tumor." Consistent with expectations, signatures previously correlated with enhanced responses to immunotherapy were significantly overrepresented in the low-CMDPTMS cohort (Fig 6D). GSEA revealed that the high-CMDPTMS group was significantly enriched in cytochrome P450-mediated drug metabolism pathways and xenobiotic metabolic processes (S5 Fig A in S1 File). Furthermore, GSEA-KEGG revealed the five most prominently enriched pathways across the CMDPTMS subgroups. Notably, focal adhesion signaling was markedly overexpressed in the high CMDPTMS group but significantly suppressed in the low CMDPTMS group, implying distinct immunotherapeutic susceptibility patterns between subgroups (S5 Fig B, C in S1 File)

Tumor mutational burden is a biomarker used to predict tumor response to immune checkpoint inhibitors. Consequently, we evaluated their distributions across the two CMDPTMS subgroups. The low-CMDPTMS cohort displayed elevated TMB levels, indicating greater immunogenicity (Fig 6E). Survival analysis additionally indicated that CMDPTMS could act as a supplementary parameter in conjunction with TMB to categorize patient prognoses (Fig 6F). Importantly, individuals with high CMDPTMS combined with high TMB tended to have improved survival outcomes.

## Predicting immunotherapy efficacy and identification of potential drugs

To thoroughly assess the function of CMDPTMS within the context of OS immunotherapy, a methodical analysis was performed. First, we examined the IMvigor-bladder cancer cohort, leveraging its extensive prognostic and treatment-related data as a common external validation cohort for the prognostic model [34]. The findings indicated that immunotherapy manifested delayed clinical impacts, as demonstrated by survival variations evident after a three-month treatment period (S6 Fig A, B in S1 File). The group with lower CMDPTMS values had superior prognostic results, suggesting a greater benefit from immunotherapy in the external validation cohort. Furthermore, the distribution of CMDPTMS differed among response categories, with the responder category (complete response) presenting notably lower scores than the non-responder category (progressive disease) (S6 Fig C in S1 File).

To explore potential biological mechanisms underlying CMDPTMS, we calculated TIP score. As anticipated, the group with low CMDPTMS displayed marked differences, especially at steps 4 (B cell recruiting) and 6, which aligns with our previous findings (S6 Fig D in S1 File). Additionally, TIDE algorithm suggested that the low-CMDPTMS group had a greater likelihood of responding favorably to immunotherapy (S6 Fig E in S1 File). Subclass mapping involving melanoma patients undergoing immunotherapy further confirmed that reduced CMDPTMS levels were linked to a more effective response to PD-1 therapy (S6 Fig F in S1 File). Validation across multiple immunotherapy cohorts revealed that a low CMDPTMS was correlated with improved immunotherapy outcomes and post-treatment prognosis (S6 Fig G-I in S1 File).

## Therapeutic response prediction

We evaluated the sensitivity of common anti-tumor drugs across CMDPTMS groups. Our results indicated that low CMDPTMS was associated with sensitivity to 30 drugs, whereas high CMDPTMS correlated with sensitivity to 7 different drugs (Fig 7A). Subsequently, potential therapeutic agents for OS were identified by analyzing the CTRP and PRISM-derived drug response datasets. Ten compounds—barasertib, BMS−986020, D−4476, GSK2110183, GZD824, masitinib, norfloxacin, ponatinib, SR−27897, and TG100−115—were identified from the PRISM cohort (Fig 7B), whereas

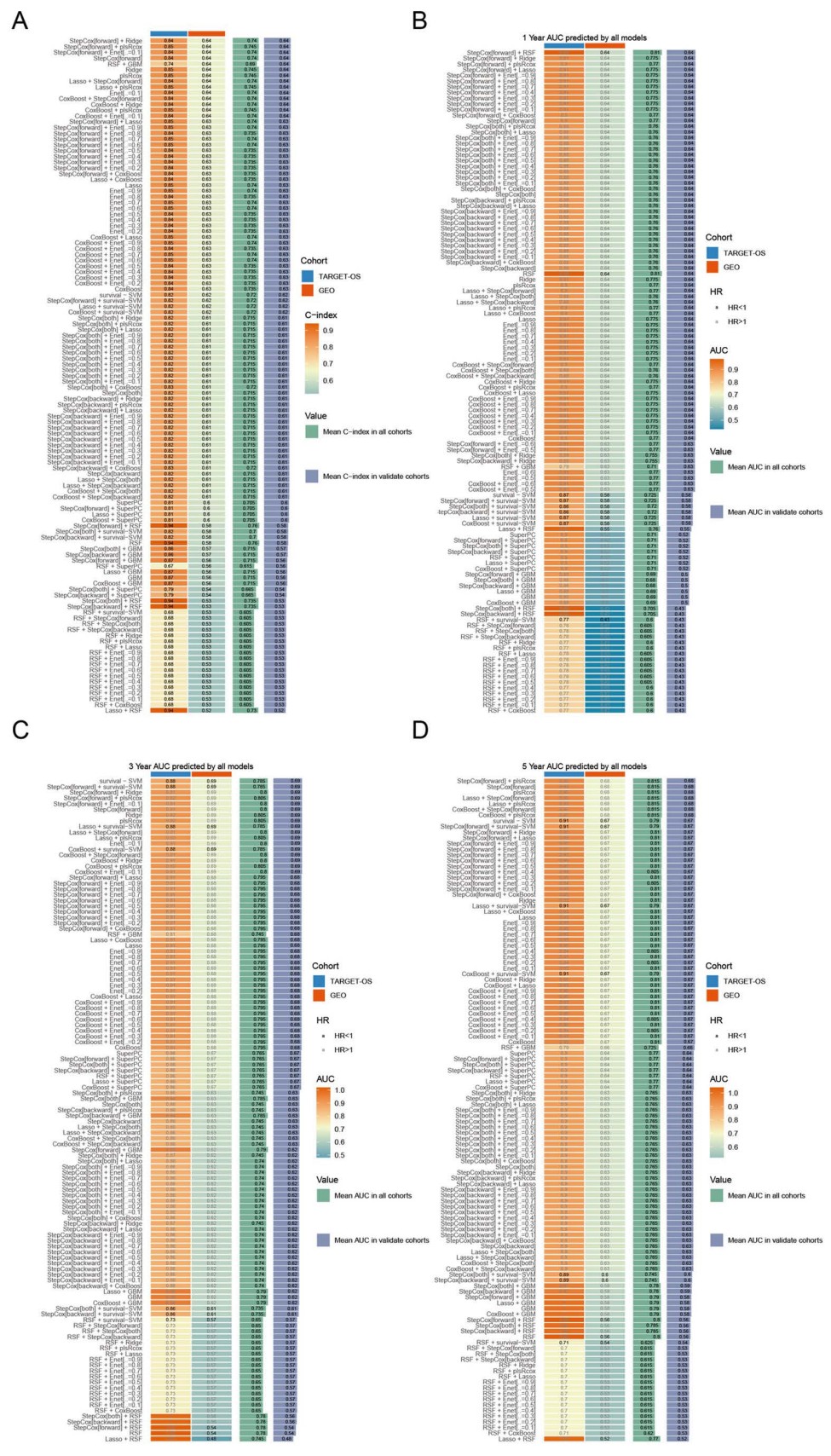

**Fig 4. Constructing and validating consensus machine learning-derived post-translational modification gene signature (CMDPTMS) in OS.** (A) C-index of each model across cohorts, ranked by average C-index in validation cohorts. (B-D) AUC values for 1-year, 3-year, and 5-year predictions across cohorts, ranked by average AUC in validation cohorts.

ten additional compounds—axitinib, AZD8055, barasertib, bleomycin A2, BRD − K97651142, cucurbitacin I, doxorubicin, etoposide, PI − 103, and tretinoin—were identified from the CTRP cohort (Fig 7C). These results underscore the potential utility of CMDPTMS in guiding personalized treatment strategies for OS patients.

## Multi-omics validation of hub genes in OS

We analyzed core gene expression profiles of CMDPTMS across seven distinct cell types and visualized the results using a heatmap. The analysis demonstrated that VIM exhibited predominant expression across 5 distinct cell types relative to other core genes (Fig 8A). UMAP plots further illustrated the distribution of core genes among diverse cell populations (Fig 8B). ST analysis corroborated these findings, demonstrating that, compared with other core genes, VIM was predominantly expressed in six cell types (Fig 8C). Notably, cells with low expressions of UBE3B, USP51 and VIM were mainly observed at the early stages of the differentiation pathway, whereas those with high expression were concentrated at later differentiation stages (S7 Fig A, B in S1 File). Additionally, cells with high PPP1CC, VPS25, and VIM expressions were primarily found at the late differentiation stages in OS, as confirmed by ST analysis (S8 Fig A, B in S1 File).

To further corroborate these observations, we examined VIM expression at both mRNA and protein levels using the Human Protein Atlas (HPA) database and RT-qPCR, given that VIM exhibited peak expression during the late differentiation stages of pseudotime analysis in both scRNA-seq and ST of OS. Subsequent analysis of the U2-OS cell lines through fluorescence staining suggested cytoplasmic localization of VIM (S9 Fig A in S1 File). RT-qPCR was used to measure *VIM* mRNA expression levels in MG-63, U-2 OS, and a normal osteoblast cell line (hFOB1.19) (S9 Fig B in S1 File). The data demonstrated a significant increase in *VIM* expression in osteosarcoma cells compared with that in osteoblast hFOB1.19 cells. To examine the role of *VIM*, we inhibited its expression in MG-63/U-2 OS cells using targeted siRNA, and knockdown efficiency was verified through RT-qPCR analysis (S9 Fig C, D in S1 File). Subsequently, CCK-8 assays demonstrated that silencing *VIM* decreased the proliferative capacity of MG-63/U-2 OS cells (S9 Fig E, F in S1 File). Furthermore, migration was examined by a Transwell assay, and VIM silencing significantly abrogated the migration of cells (S9 Fig G, I in S1 File). These findings suggest that VIM may facilitate tumorigenesis in OS, although the exact mechanism requires further exploration.

## Discussion

Recent advances in molecular biology have clarified the vital role of PTMs in orchestrating tumorigenesis and immune modulation in OS. Glycosylation, a fundamental PTMs, has been demonstrated to critically regulate chondrocyte and osteoclast differentiation processes, which are intrinsically linked to progression of OS [35]. Ubiquitination, another prominent PTMs, serves as a key regulatory mechanism in protein degradation and signal transduction pathways, contributing significantly to the pathogenesis of OS through its involvement in various physiological and pathological processes [36]. Furthermore, emerging evidence suggests that the dysregulation of SUMOylation and NEDDylation profoundly impacts multiple facets of tumor evolution and neoplastic transformation, including hypoxia adaptation, tumor suppressor expression, oncogenic mediator activity, and drug resistance mechanisms in OS [12]. Despite these significant findings, investigations of PTMs at single-cell resolution in OS remain largely unexplored. To address this critical gap in knowledge, our study employed an integrative multi-omics approach that combines scRNA-seq, ST, and bulk RNA-seq data to comprehensively elucidate the malignant characteristics of PTMs at both single-cell and spatial resolution levels.

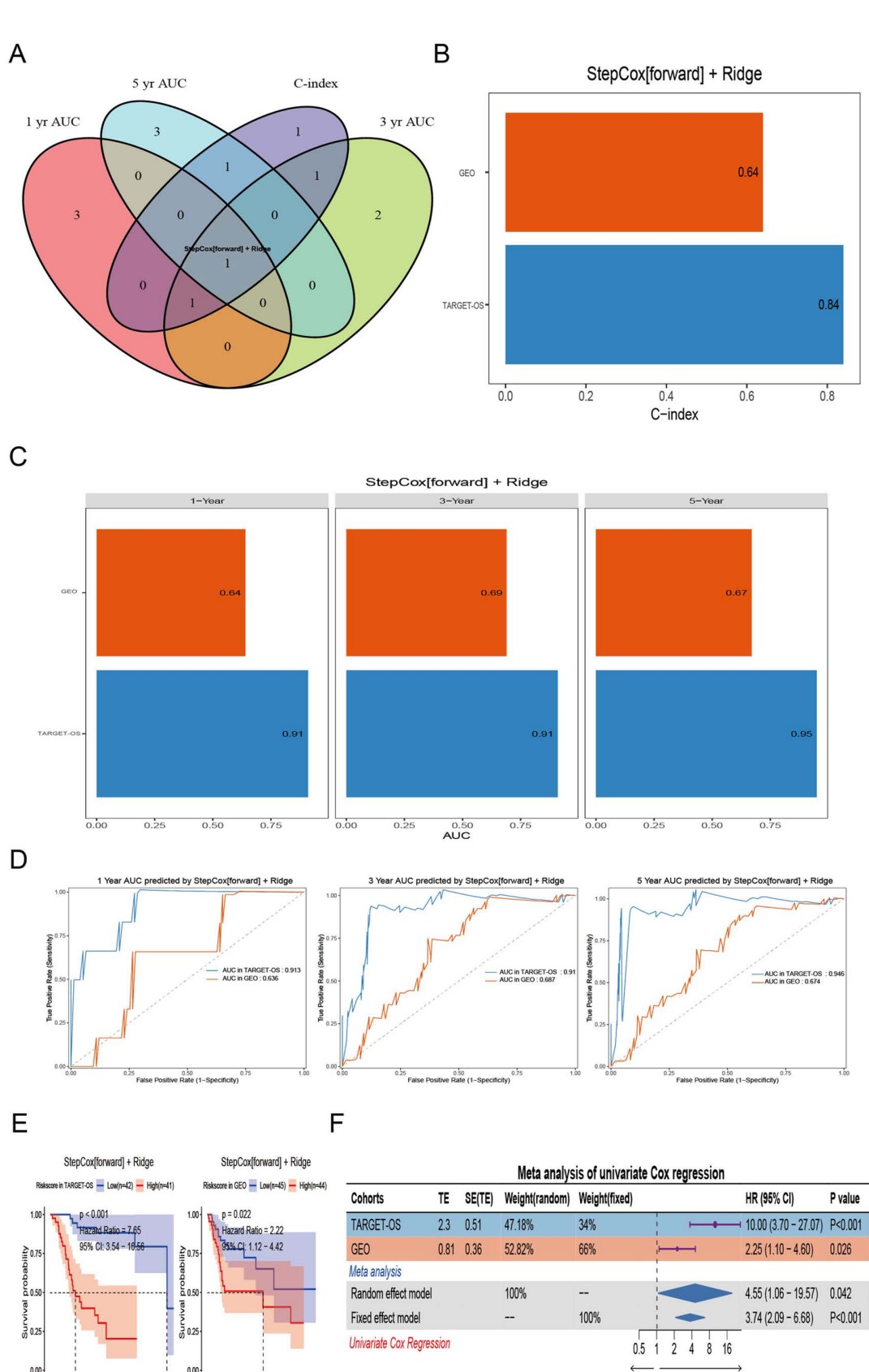

**Fig 5. Evaluation of prognostic model performance.** (A) Venn diagram showing overlapping features among the top five prognostic models based on C-index and AUC values at 1-, 3-, and 5-year intervals. (B) C-index distribution of the StepCox[forward] + Ridge survival model across cohorts. (C) Temporal discrimination accuracy of the model assessed via 1-, 3-, and 5-year AUC values. (D) ROC curves at 1-, 3-, and 5-year intervals evaluating predictive performance in distinct datasets.

Using scRNA-seq and ST, we mapped the cellular and spatial immune landscape of PTMs in OS. Our findings revealed that PTMs are heightened in tumor osteoblastic cells, with PTMshighos cells exhibiting a terminally differentiated state and an increased metastatic potential. A recent study reported that osteoblastic cells are more prevalent in OS tumor tissues, whereas NKT cells and T cells are primarily found in lymph nodes, a finding that is consistent with our results [18]. Furthermore, the highly expressed PTMs observed in our study were associated with reduced interactions between osteoblastic and immune cells, including B and T. These findings suggest that PTMs within the TME may suppress antigen presentation and weaken antitumor responses [37,38]. Therapeutic strategies targeting OS should prioritize the modulation of the immunosuppressive TME, as converting "cold" immunosuppressive tumors into "hot" immune-activating tumors represents a promising approach to increase OS treatment efficacy [39]. Evidence suggests that dysregulated PTMs in OS cells facilitate immune evasion through diverse mechanisms [40,41]. Using the MISTy algorithm, we confirmed that PTMshighos cells frequently colocalize with fibroblasts and T cells in OS. Notably, cancer-associated fibroblasts have been identified as pivotal contributors to immunosuppression within the TME [42]. Concurrently, cytotoxic CD8+T cells play a critical role in orchestrating anti-tumor immune responses; however, their functionality is significantly impaired by exhaustion [43–46], thus fostering tumor progression and reinforcing immune suppression. These findings suggest that these cells, in conjunction with PTMshighos cells, may orchestrate immune evasion mechanisms within the TME.

Given the dismal survival outcomes observed in OS patients, the identification of critical prognostic factors has become a pressing clinical priority [47]. To address this challenge, we conducted a comprehensive evaluation of 117 machine-learning algorithms, ultimately developing the CMDPTMS on basis of the expression profiles of ten pivotal PTMs. The CMDPTMS exhibited superior prognostic performance, significantly surpassing traditional clinical parameters, including tumor stage, in predicting overall survival outcomes. Our findings revealed an elevation in VIM levels in OS cells, which correlated with increased cell proliferation and migration, aligning with prior studies that reported markedly increased levels of VIM and CD63 in plasma exosomes from OS patients relative to those from healthy individuals [48]. The critical role of PTMs in cancer progression has been increasingly recognized across various malignancies. For instance, FZD8, a gene implicated in cancer development and stem cell reprogramming, undergoes progressive silencing during tumorigenesis [49]. In nasopharyngeal carcinoma, elevated PPP1CC expression facilitates non-homologous end joining-mediated DNA repair, contributing to radioresistance and an unfavorable prognosis [50]. VPS25 has been identified as a promoter of immunosuppressive conditions in head and neck squamous cell carcinoma [51], whereas BAG2 facilitates gastric cancer growth and spread via the ERK1/2 pathway [52]. The control of TRIM35 ubiquitination over PKM2 tetramer and dimer expression impacts breast cancer severity by altering the Warburg effect [53]. UBE3B plays a role in advancing breast cancer by inhibiting HIF-2α degradation [54], and USP51 increases colorectal cancer stemness and resistance to chemotherapy through a positive feedback loop with HIF1A [55]. Furthermore, BRD4 inhibition has been demonstrated to increase cervical cancer radiosensitivity by attenuating DNA repair mechanisms [56].

To elucidate the immune landscape associated with CMDPTMS subgroups, we conducted comprehensive immune-related signature enrichment analysis using the IOBR R package. Our findings revealed that, compared with the high-CMDPTMS cohort, the low-CMDPTMS cohort presented significantly greater TMB levels, greater levels of immunosuppressive molecular markers, and antitumor immunity [57]. Through the application of predictive algorithms including TIDE, TIP, and subclass mapping, we consistently observed superior immunotherapy responsiveness in the low-CMDPTMS subgroup. Moreover, the KEGG focal adhesion signaling pathway, which is inversely correlated with PD-1/PD-L1 blockade

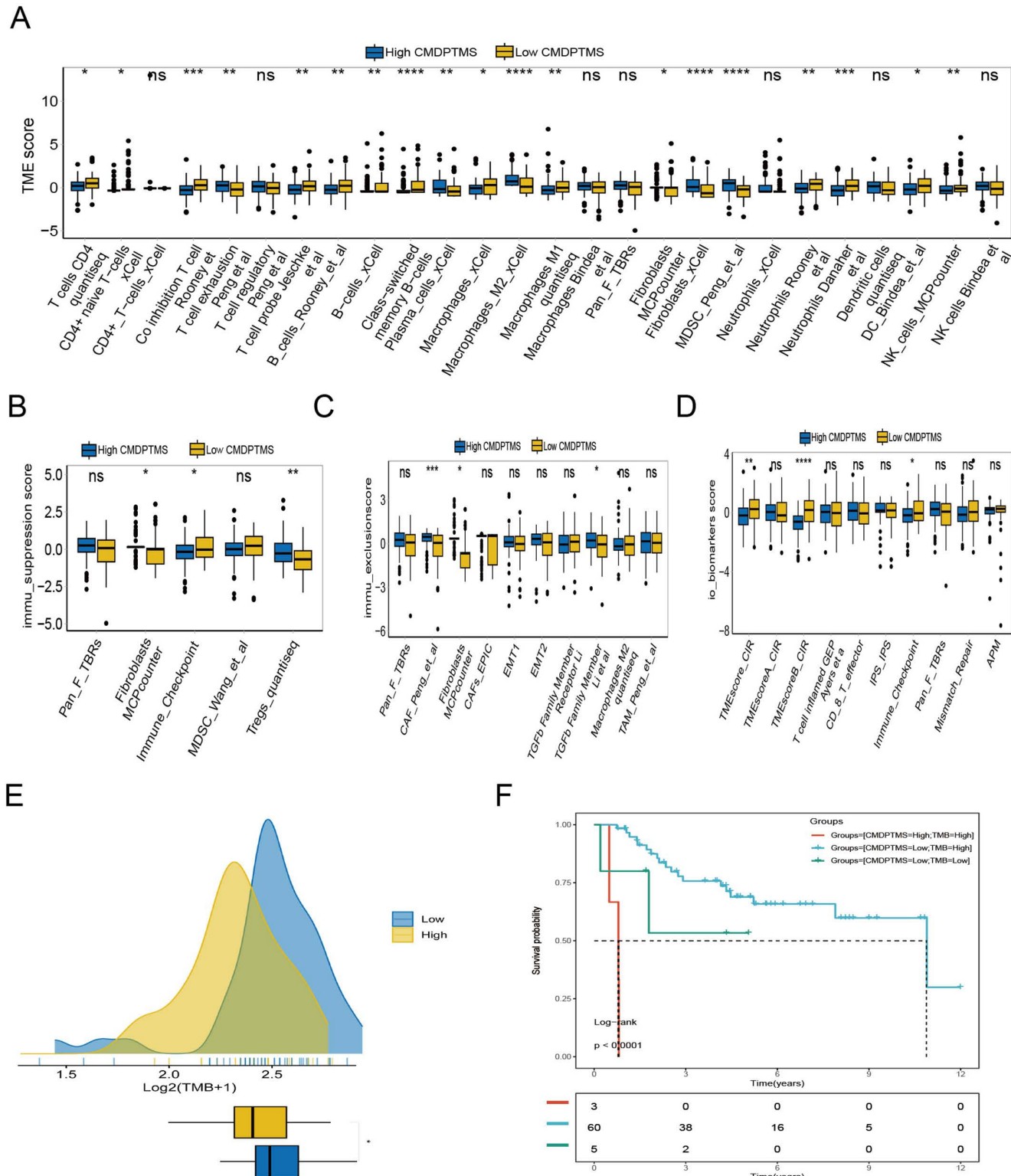

**Fig 6. Tumor microenvironment (TME)-related molecular characteristics in high- and low-CMDPTMS patients.** (A) Comparison of TME immune cell type signatures. (B) Distribution of immune suppression signatures. (C) Distribution of immune exclusion signatures. (D) Comparative analysis of immunotherapy biomarkers. (E, F) Differences in tumor mutational burden (TMB). (G, H) Survival analysis integrating CMDPTMS with TMB.

A

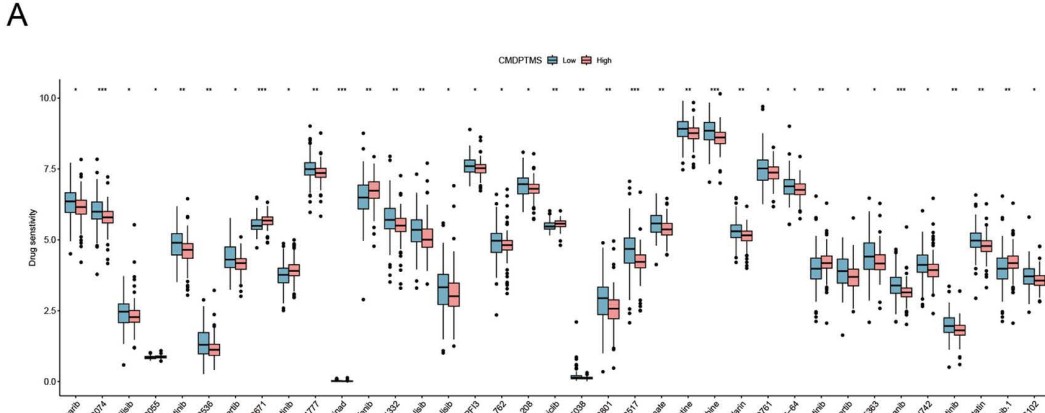

B

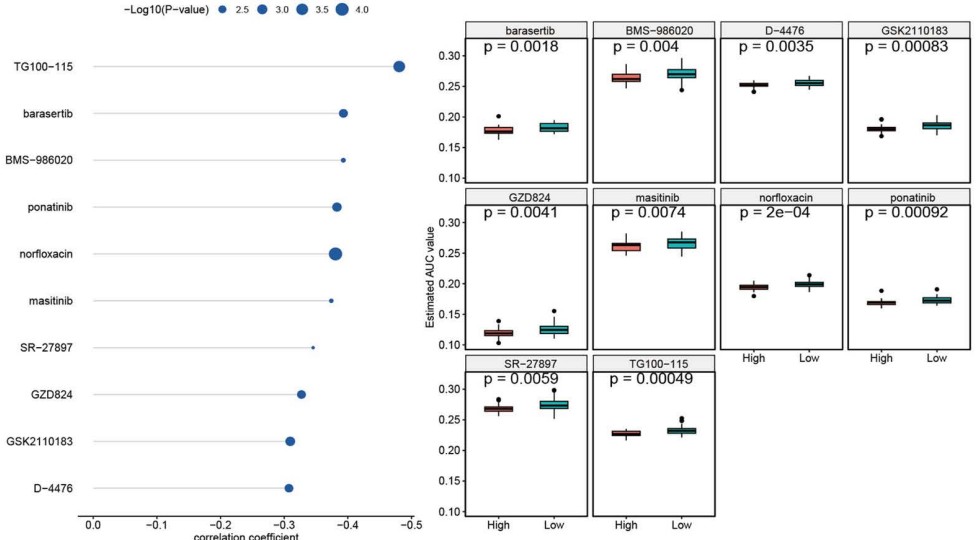

C

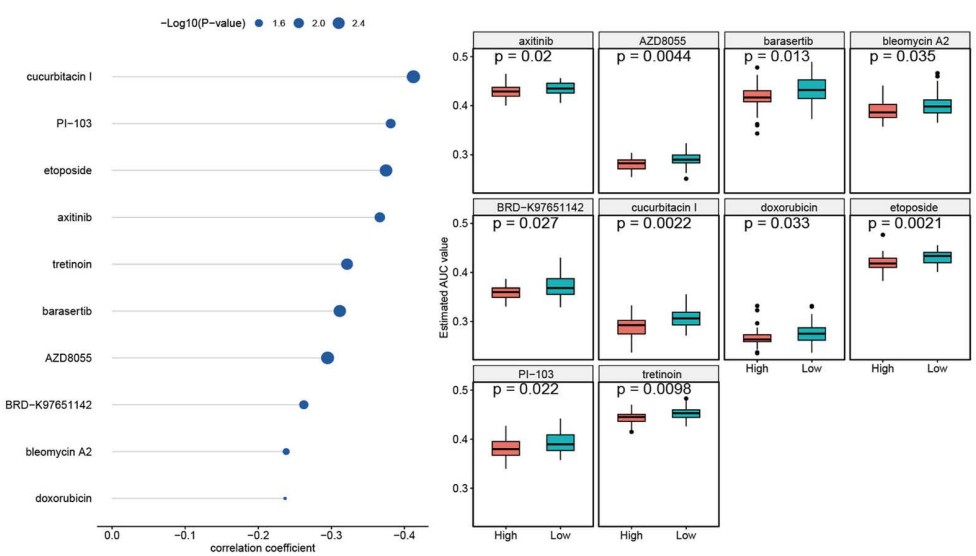

**Fig 7. Therapeutic response prediction.** (A) Correlation between CMDPTMS and therapeutic response prediction. (B, C) Correlation and differential analysis of drug sensitivity for potential drugs screened from PRISM and CTRP datasets.

sensitivity, was markedly upregulated in the high CMDPTMS group, consistent with previous reports [58]. These results suggest that CMDPTMS serves as a valuable tool for the early stratification of immunotherapy-sensitive patients. In light of the limited immunotherapy efficacy observed in the high-CMDPTMS population compared with the low-CMDPTMS population, we performed an extensive screening of potential therapeutic agents, including AZD8055, erlotinib, PF-4708671, dabrafenib, ribociclib, ulixertinib and ulixertinib.1. Notably, AZD8055 has demonstrated potent antitumor activity in colon cancer cells through dual inhibition of mTOR signaling and cell-cycle progression [59]. PF4708671 enhances tamoxifen-mediated cytotoxicity in MCF-7 breast cancer cells via translational suppression of survivin [60]. Erlotinib has been evaluated as a neoadjuvant treatment for non-small cell lung cancer and has shown effectiveness in a phase II, single-arm, prospective study [61]. Erlotinib demonstrated clinical efficacy as a neoadjuvant therapeutic agent in osteosarcoma patients during initial phase I clinical trials [62]. Dabrafenib has been reported to inhibit Egr-1-mediated adhesion of thyroid cancer cells to pulmonary microvascular endothelium, potentially reducing metastatic potential [63]. Dabrafenib enhances chemosensitivity in methotrexate-resistant U2-OS osteosarcoma cells [64]. Furthermore, ribociclib has been investigated in conjunction with platinum-based chemotherapy in ovarian cancer patients in a phase I clinical trial [65]. Additionally, the use of ulixertinib in combination with gemcitabine and nab-paclitaxel for the treatment of metastatic pancreatic adenocarcinoma has been examined in a phase Ib clinical trial [66].

Although these results are encouraging, the research presents certain constraints. The small sample size restricts generalizability, and the focus on PTMshighos cells in OS limits insights into other tumor types. Initial pharmacological screening revealed that 10 compounds exhibited distinct response profiles between CMDPTMS-defined patient subgroups; however, comprehensive clinical validation remains imperative. The mechanisms underlying CMDPTMS gene activity require further exploration, and its practical clinical relevance must be confirmed through extensive, multicenter studies.

## Conclusions

In summary, we integrated scRNA-seq, ST data, and TCGA bulk RNA-seq cohorts to dissect various cell types and highlight the role of PTMsos cells in OS. A machine learning-derived CMDPTMS demonstrated robust prognostic and immunotherapy response prediction capabilities. High-CMDPTMS groups exhibited poor prognosis and low immunotherapy responses, prompting investigation into therapeutic benefits of AZD8055, PF−4708671, erlotinib, dabrafenib, ribociclib, and ulixertinib. This study, leveraging multi-omics data and computational algorithms, establishes a framework for early detection and tailored therapeutic strategies for OS.

## Supporting information

**S1 File.** S1 Table. Post-translational modifications-related genes list. S2 Table: Sequences for siRNA. S3 Table: RT-qPCR primer sequences. S1 Fig. Gene set enrichment analysis (GSEA) based on dysregulated genes of the osteoblastic cells. (A, B) GSEA analysis of dysregulated genes in osteoblastic cells based on PTMshighos and PTMslowos. S2 Fig. Intercellular communication analyses. (A) Intercellular communication pathways among PTMs-associated osteoblastic cells. (B) Correlation circle plots illustrating interactions between different cell types. (C) Heatmap showing cell–cell ligand-receptor correlations. (D) Sankey diagram visualizing ligand-receptor-transcription factor interactions in fibroblasts with higher PTMs expression. S3 Fig. Clinical practice value of CMDPTMS. (A, B) Comparison of CMDPTMS with 16 other published models in the TARGET-OS and GEO-OS cohorts. S4 Fig. Prognostic significance of CMDPTMS. (A) Nomogram predicting 1-, 3-, and 5-year overall survival in OS patients. (B) Kaplan-Meier survival analysis comparing

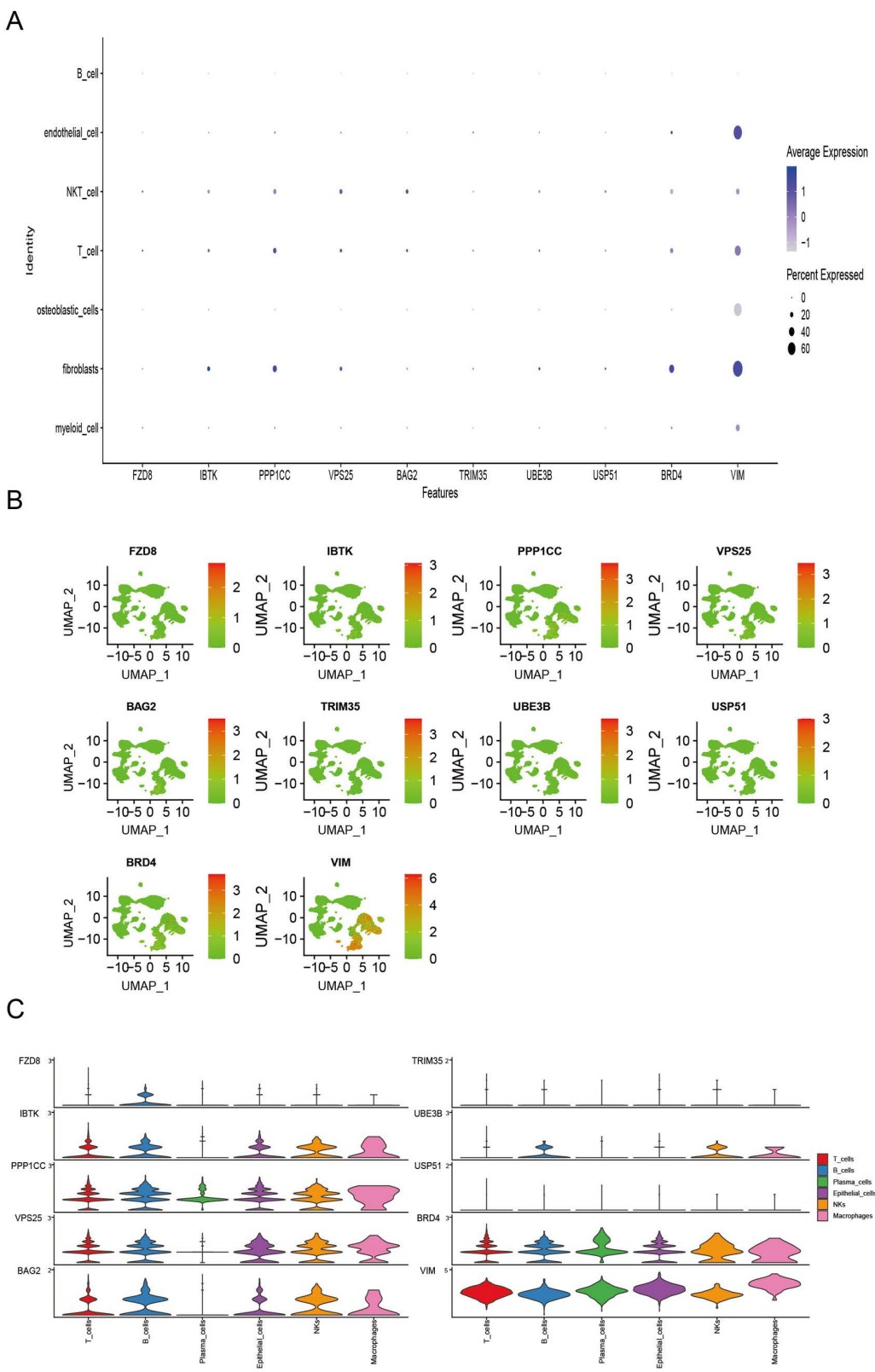

**Fig 8. ScRNA-seq and ST profiles of core genes in OS.** (A) Bubble plots depicting core gene expression across cell types in scRNA-seq data. (B) UMAP plots illustrate core gene distribution and expression in different cell types. (C) Violin plots showing core gene expression across cell types in ST data.

high- and low-score groups. (C-E) Decision curve analysis assessing the clinical utility of the nomogram for 1-, 3-, and 5-year survival predictions. S5 Fig. GSEA-KEGG pathway enrichment analysis in the low CMDPTMS and high CMDPTMS groups. A. GSEA-KEGG analysis of metabolic-related pathways in the low CMDPTMS and high CMDPTMS groups. (B, C). GSEA-KEGG analysis in different CMDPTMS groups. S6 Fig. The predictive value of CMDPTMS in immunotherapy response among OS patients. (A) Differences in restricted mean survival (RMS) time at 6 months and 1-year post-treatment. (B) Differences in long-term survival (LTS) at 3 months post-treatment. (C) Distribution of CMDPTMS across immunotherapy response groups. (D) Variations in TIP process activation levels. (E) Immunotherapy response predictions based on the TIDE algorithm. (F) Immunotherapy response predictions based on the subclass mapping algorithm. (G) Distribution of CMDPTMS across immunotherapy response groups in the GSE91061 dataset. (H, I) Survival analysis in the GSE78220 and GSE135222 datasets. S7 Fig. Pseudotime analysis of core genes of CMDPTMS in scRNA-seq data. (A, B) Core genes were divided into three evolutionary branches. S8 Fig. Pseudotime analysis of core genes of CMDPTMS in ST data. (A, B) Core genes were divided into four evolutionary branches. S9 Fig. Validation of VIM role in OS. (A) Cellular localization of VIM in U2-OS cell lines. (B) qRT-PCR analysis of *VIM* expression in normal osteoblast and OS cell lines. (C, D) The level of *VIM* transfection with siRNA was analyzed by qRT-PCR. (E, F) CCK-8 assays were performed to examine the potential proliferation of siVIM cells or negative control cells. (G, H) Transwell assays were performed to assess the migratory potential of siVIM cells and negative control cells.
(ZIP)

## Author contributions

**Conceptualization:** Yue Cui, tao Ding.

**Software:** Yue Wu.

**Supervision:** tao Ding.

**Writing – original draft:** Yue Cui.

**Writing – review & editing:** Dongyu Jiang, tao Ding.

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
