## [Decision Letter · Decision Letter 0]

19 Jun 2025

PONE-D-25-25008Single-cell and spatial transcriptomics reveal post-translational modifications in osteosarcoma progression and tumor microenvironmentPLOS ONE

Dear Dr. Ding,

Thank you for submitting your manuscript to PLOS ONE. After careful consideration, we feel that it has merit but does not fully meet PLOS ONE’s publication criteria as it currently stands. Therefore, we invite you to submit a revised version of the manuscript that addresses the points raised during the review process.

We look forward to receiving your revised manuscript.

Kind regards,

Yisheng Chen

Academic Editor

PLOS ONE

Journal Requirements:

5. Please upload a copy of Supporting Information Figure/Table/etc. S1 to S3 table which you refer to in your text on page 5 and 8.

Reviewers' comments:

Reviewer's Responses to Questions

**Comments to the Author**

1. Is the manuscript technically sound, and do the data support the conclusions?

Reviewer #1: Yes

Reviewer #2: Partly

Reviewer #3: Yes

Reviewer #4: Yes

2. Has the statistical analysis been performed appropriately and rigorously? 

Reviewer #1: Yes

Reviewer #2: I Don't Know

Reviewer #3: Yes

Reviewer #4: Yes

3. Have the authors made all data underlying the findings in their manuscript fully available?

Reviewer #1: Yes

Reviewer #2: Yes

Reviewer #3: Yes

Reviewer #4: Yes

4. Is the manuscript presented in an intelligible fashion and written in standard English?

Reviewer #1: Yes

Reviewer #2: No

Reviewer #3: Yes

Reviewer #4: Yes

5. Review Comments to the Author

Reviewer #1: Dear author,

It is a great honor for me to serve as a reviewer and participate in the review of this article. I will offer some targeted review comments to help improve the research content and the quality of the paper.

1.Although the role of PTM in osteosarcoma is not fully clarified in the text, it lacks detailed comparisons with existing similar studies and fails to clearly highlight the unique innovation points of this study in the field of the association between PTM and osteosarcoma. It is suggested to supplement.

2. It only explains the classification of tumor cells based on PTM scores using the AddModuleScore method, without elaborating in detail on the construction basis of the scoring system and the verification of its rationality. Further explanation is needed.

3. The application of the RCTD method to map single-cell groups to spatial transcriptomes lacks an assessment of the applicability and accuracy of this method. Relevant verification experiments or literature evidence should be supplemented.

4. The prognostic model was constructed using bulk transcriptome data. The data screening criteria and statistical methods for model construction were not clearly defined, which may affect the universality and reliability of the model. Detailed explanations are required.

5. Although different subtypes of tumor cells have been identified, the specific biological functions of these subtypes in the occurrence, development and metastasis of osteosarcoma have not been deeply explored. It is suggested to supplement the mechanism research.

6. Ten hub genes were determined to form CMDPTMS. The screening process and criteria were not elaborated in detail. Statistical analysis and biological basis for gene screening should be supplemented.

7. It is emphasized that CMDPTMS has a strong prognostic prediction ability, but it is only verified based on existing data and lacks verification from external independent datasets. Supplementary verification experiments are needed to enhance persuasiveness.

8. It was pointed out that the high CMDPTMS group had a decreased response to immunotherapy, but the intrinsic molecular mechanism was not deeply explored. It was suggested to conduct an analysis from the perspectives of signaling pathways and immune cell infiltration.

Nine. Seven drugs were proposed as potential treatment options, but there is a lack of association analysis between the drugs and the mechanism of action of CMDPTMS or osteosarcoma cells. Relevant experimental data or theoretical basis need to be supplemented.

10. It was found that the PTM highos phenotype has a close interaction with fibroblasts. The specific molecular mechanism of the interaction and its influence on the development of osteosarcoma have not been deeply studied. It is recommended to supplement.

11. The sources of the samples, inclusion and exclusion criteria in the study were not clearly defined, and there may be issues with the representativeness of the samples. Detailed explanations are needed to assess the universality of the research results.

12. The references are not sufficient. It is suggested to supplement the following references:39445224�37435067�37040518�35101942�35140113�37251708�38319150�38943472

Reviewer #2: This manuscript addressed an important and timely topic, but the writing is highly specialized and the readability isn't the best for a physician scientist like me. It should be understood by clinicians so the writing should be more clear, and they should avoid so much scientific jargon.

The topic of the manuscript is highly relevant, however the language in the manuscript can be more straightforward to be understood by other researchers and clinicians in this field. Here's my recommendations:

Would strongly recommend proofreading the manuscript with an English language expert as there're multiple phrases in this manuscript that don't seem very soundly, for example line 261: TMB was widely recognized biomarkers for predicting patient responses...", and many others. I would suggest asking a local physician who sees osteosarcoma patients to look at this manuscript and work with the authors to make the text easier to understand to the medical community who is not an ultra expert in scRNAseq or ST. This will make this manuscript very helpful and clinically applicable.

Line 16: why is a more adequate characterization of post-translational modifications in OS important? please state briefly the evidence prior to this manuscript that this plays a role in outcomes or management decisions like survival, recurrence or resistance to therapy. This will give context to the reader.

Line 18: please spell out RCTD and scRNA-seq. They're spelled out later in the manuscript but they should be spelled out here as well.

Line 54: spell out small ubiquitin-like modifier (SUMO)

Line 61: the authors use several times the word "intriguingly", however it's not clear why it's intriguing. Is it because the data would point to a different result? I would clarify why. If the word is just used as an adverb to continue mentioning the data, I would recommend using a different one, like "furthermore", "of note", etc., or simply state the data.

Line 101, 105, 108, 113, 132 and 139: please reference the source of the Seurat R package, the Monocle package, MistyR package, the limma package, the IOBR package and TIDE and TIP.

Line 149: can you state what was the purpose of transiently suppressing VIM expression? This is partially addressed in the paragraph of lines 306-326, but I would state the rationale in line 149 to guide the reader. I would also state what suggested the researchers to look at VIM or what is the preexisting data that suggested that VIM should be evaluated because it plays a role in OS tumorogenesis.

Line 171: were the T cells or could the T cells be separated between T4 and T8? later in discussion the authors mention the role of T8+ cells playing a role in TME and it would be nice to see the data analysis looking at this subgroup of T cells in the study. If the T4 vs. 8 can not be separated in the analysis, this should be stated as the reader will have this question and it will be helpful to know that the authors addressed this or there was a reason why it could not be done.

Paragraph lines 229-239: the information seems difficult to understand; what is the conclusion of this information, and what is the relationship with the last phrase "Moreover, High- CMDPTMS...". This paragraph should be modified and be made more clear.

Line 261: the phrase "TMB was widely recognized biomarkers for predicting patient responses..." doesn't seem soundly.

Line 267: for the phrase "CMDPTMS combined with high TMB showed a tendency toward improved survival outcomes", can the data be shown graphically? given that the quality of the images isn't good to read, it's unclear if the data is shown there, and it should be referenced in parenthesis. TMB should be spelled out, and perhaps changing it for something like "Tumor mutational burden is a biomarker used to predict tumor response to immune checkpoint inhibitors".

Line 275: please state what specific "benefit" was seen with immunotherapy in the lower CMPTMS group, was it survival or response to therapy or what other parameter? Can by how much or what is the data supporting this statement. This paragraph should also be evaluated or proofread, as it seems difficult to follow through; for example the phrase on line 272 seems tangential and it's not clear what are the "finings" of the analysis, was it that survival was higher at 3 months of immunotherapy in the lower CDMPTMS group? the phrases don't seem clear and the reader gets confused.

Line 285: why are melanoma patients being analyzed? the manuscript is on osteosarcoma, so why is melanoma included here now? this should be described, the rationale and the relationship with the analysis on OS.

IMAGES: many of the small font of the images can't be read easily, so I would make sure the quality of the images is optimized (e.g. fig 2 G I J and many other images).

IMAGES: some of the images probably can't have a more clear font, so I would suggest modifying the images perhaps cutting some of the information and leave the complete images bigger and on their own in a supplemental document.

Reviewer #3: The manuscript uses a combined multi-omics method to explain how post-translational modifications (PTMs) affect osteosarcoma (OS). The authors use advanced tools like single-cell RNA sequencing (scRNA-seq) and spatial transcriptomics (ST) to help us understand how osteosarcoma (OS) grows, how the immune system reacts, and what treatment options might be available. The development of a new gene signature based on machine learning for post-translational modifications (CMDPTMS) is a fresh way to predict how patients will do and respond to treatment. However, several areas of the manuscript would benefit from clarification or further elaboration. The following comments are provided for the authors' consideration.

1.The introduction highlights PTMs as crucial elements in both osteosarcoma development and cancer progression. A clearer presentation with structured information and clear distinctions between different PTM types like acetylation and methylation would improve comprehension. The discussion introduces advanced technologies such as scRNA-seq and ST yet fails to clearly articulate the existing knowledge gap about PTMs in osteosarcoma.

2.The methodology contains enough detail but needs more explicit descriptions in certain sections. The analytical clarity of "AddModuleScore" and "MistyR" would improve if authors detailed their procedural steps and parameters. Reproducibility would benefit from additional details about quality control measures and filtering standards used in scRNA-seq and ST data. The CMDPTMS development description requires additional explanation on why particular machine learning algorithms were chosen and more thorough details about their validation process.

3.The findings presented in this study show comprehensive clarity and provide definitive support for using scRNA-seq and ST methods to investigate PTMs in osteosarcoma. Nonetheless, several areas could be further developed. Figure 2 presents significant pseudotemporal shifts in PTM expression among cell types which require further analysis to understand their biological significance. Figure 6 presents an informative immune landscape analysis which would benefit from a detailed description of how CMDPTMS subgroups relate to immune responses to enhance interpretation. The authors should provide more details about the effects of PTM modifications on immune system interactions and their application in therapeutic methods. The drug sensitivity analysis in Figure 7 provides useful insights but would benefit from additional information regarding the choice of the 10 drugs and their clinical importance in osteosarcoma treatment.

4.The authors applied which methods to address batch effects between scRNA-seq and ST data in their study? Did the authors utilize any technique like kBET or LISI to evaluate cell cluster stability throughout the data integration process? Inclusion of this information would be advantageous.

5.For insights into gene signatures related to tumor progression and immune responses, I recommend the following articles [Front. Aging Neurosci. 2022;14:951197] and [Bioact. Mater. 2024;17:206]. These articles may offer valuable perspectives for your ongoing research into osteosarcoma and the role of post-translational modifications in cancer progression.

6.The study primarily uses computational methods without experimental confirmation of PTM-driving genes VIM, FZD8 and PPP1CC yet it needs experimental evidence from existing research about their involvement in OS or future validation approaches.

7.Methods did not detail patient informed consent, ethics approval number, or sample collection period.For children's OS, ethical requirements are stricter, and it is recommended to provide corresponding explanations.

Reviewer #4: Osteosarcoma is a highly aggressive malignant bone tumor that mainly affects children and young people, and has a poor prognosis. This study focuses on the role of post - translational modifications (PTMs) in osteosarcoma progression and the tumor microenvironment, which is of great clinical significance. By integrating single - cell RNA sequencing (scRNA - seq), spatial transcriptomics (ST), and machine learning methods, it reveals the mechanisms of PTMs in osteosarcoma and constructs the CMDPTMS prognostic model, offering new perspectives and potential biomarkers for precise osteosarcoma treatment and prognosis prediction.

The authors obtained high - quality single - cell sequencing and spatial transcriptomics data from multiple public databases and used appropriate algorithms for data preprocessing, standardization, and dimensionality reduction. However, the authors didn't elaborate enough on the parameter selection and basis for some key steps in data processing, such as the specific basis of parameter settings and their impact on the results.

The authors built a prognostic model through univariate Cox regression analysis and various machine learning algorithms and evaluated its performance. The model validation was relatively adequate, but the model - building process was complex, involving multiple algorithm combinations and parameter selections. The authors didn't detail how the final model architecture and parameter settings were determined, nor did they address the risk of overfitting.

In the discussion section, the authors conducted an in - depth discussion on the research results, explored the mechanisms of PTMs in osteosarcoma development in combination with existing literature, and analyzed the clinical significance and potential application value of the CMDPTMS prognostic model. However, the discussion on the limitations of the research results and future research directions was relatively limited.

Despite extensive bioinformatics analysis and machine - learning model construction, the biological function verification experiments for key genes (such as VIM) were relatively preliminary, with only siRNA - mediated cell proliferation experiments. It's suggested to add more cell and animal experiments, such as cell migration and invasion assays, and evaluate the effects of VIM on osteosarcoma growth and metastasis in animal models, to further verify the mechanisms of VIM in osteosarcoma and its potential as a therapeutic target.

Although the authors integrated single - cell RNA sequencing, spatial transcriptomics, and machine learning, the depth and breadth of multi - omics data integration need to be improved. For instance, incorporating proteomics and metabolomics data could provide a more comprehensive understanding of PTMs' mechanisms in osteosarcoma and their interactions with other molecular levels, offering a more complete molecular map and therapeutic targets for precise osteosarcoma treatment.

No supplementary figures were found in the manuscript, but they were mentioned in the main text. The references cited in this article are insufficient, and in - depth comparative discussions are lacking. The background and methodology also require more literature support. Some related research should be cited�such as 10.15212/bioi-2022-0008, 10.15212/bioi-2022-0014, 10.1093/burnst/tkad020, 10.1093/burnst/tkae043, 10.34133/2022/9816272, 10.3389/fimmu.2023.1196892, 10.1111/cpr.13703 and 10.1111/cpr.13697.

6. PLOS authors have the option to publish the peer review history of their article (what does this mean? ). If published, this will include your full peer review and any attached files.

**Do you want your identity to be public for this peer review?** For information about this choice, including consent withdrawal, please see our Privacy Policy .

Reviewer #1: No

Reviewer #2: No

Reviewer #3: No

Reviewer #4: No

---

## [Author Response · Author response to Decision Letter 1]

6 Aug 2025

Editor

Dear Editor,

We are hereby submitting a revised manuscript entitled “Single-cell and spatial transcriptomics reveal post-translational modifications in osteosarcoma progression and tumor microenvironment” (Submission ID: PONE-D-25-25008) for publication. We appreciate very much for the constructive comments and suggestions from you and the reviewers. In this revised manuscript we have taken every effort to address the comments and concerns. All the changes in the text are highlighted in red. The details of our revised manuscript, as required by reviewer 1, 2, 3 and 4 as below.

Editor Comments:

Comment 1 of Editor: Please ensure that your manuscript meets PLOS ONE's style requirements, including those for file naming.

Response: Thank you for your valuable comment. We have revised our manuscript accordingly to align with the formatting and style requirements of PLOS ONE.

Comment 2 of Editor: Please note that PLOS ONE has specific guidelines on code sharing for submissions in which author-generated code underpins the findings in the manuscript.

Response: Thank you for your valuable advice. We have uploaded the code generated in this study to GitHub: https://github.com/Aa1589511336/code-of-Single-cell-and-spatial-transcriptomics-reveal-PTMs-in-osteosarcoma.

Comment 3 of Editor: We note that your Data Availability Statement is currently as follows: All relevant data are within the manuscript and its Supporting Information files

Response: Thank you for your valuable comments. We have revised our data availability statement to: “The datasets used in this study are publicly available from the TCGA database (http://cancergenome.nih.gov/) and the GEO database (https://www.ncbi.nlm.nih.gov/geo/). All relevant data are provided within the manuscript and its supporting information files.” Furthermore, we have obtained permission from the KEGG database, as attached below in the rebuttal letter.

Comment 4 of Editor: PLOS requires an ORCID iD for the corresponding author in Editorial Manager on papers submitted after December 6th, 2016. Please ensure that you have an ORCID iD and that it is validated in Editorial Manager. To do this, go to ‘Update my Information’ (in the upper left-hand corner of the main menu), and click on the Fetch/Validate link next to the ORCID field. This will take you to the ORCID site and allow you to create a new iD or authenticate a pre-existing iD in Editorial Manager.

Response: Thank you for your valuable comments. We have created an ORCID and updated my information in the main menu of the PLOS ONE submission.

Comment 5 of Editor: Please upload a copy of Supporting Information Figure/Table/etc. S1 to S3 table which you refer to in your text on page 5 and 8.

Response: Thank you for your valuable comments. We have uploaded the supplemental figures and tables along with our revised manuscript.

Reviewer(s)' Comments:

Comment 1 of Reviewer 1: Although the role of PTM in osteosarcoma is not fully clarified in the text, it lacks detailed comparisons with existing similar studies and fails to clearly highlight the unique innovation points of this study in the field of the association between PTM and osteosarcoma. It is suggested to supplement.

Response: Thank you for your valuable comments. To emphasize the novel aspects of this study, we conducted a systematic review of relevant literature published within the past five years and ultimately incorporated 16 distinct gene signatures into our analysis.

Comment 2 of Reviewer 1: It only explains the classification of tumor cells based on PTM scores using the AddModuleScore method, without elaborating in detail on the construction basis of the scoring system and the verification of its rationality. Further explanation is needed.

Response: Thank you for your valuable advice. We have added a reference to support the application of the AddModuleScore method in scRNA-seq. Furthermore, this method has been utilized in other studies, such as PMID: 40109658, PMID: 38270619, PMID: 40638251, and PMID: 40054467.

Comment 3 of Reviewer 1: The application of the RCTD method to map single-cell groups to spatial transcriptomes lacks an assessment of the applicability and accuracy of this method. Relevant verification experiments or literature evidence should be supplemented.

Response: Thank you for your valuable advice. We have added a reference to support the application of the RCTD method in mapping scRNA-seq to spatial transcriptomes.

Comment 4 of Reviewer 1: The prognostic model was constructed using bulk transcriptome data. The data screening criteria and statistical methods for model construction were not clearly defined, which may affect the universality and reliability of the model. Detailed explanations are required.

Response: Thank you for your valuable comments. We have added data screening criteria and statistical methods for model construction in the revised manuscript.

Comment 5 of Reviewer 1: Although different subtypes of tumor cells have been identified, the specific biological functions of these subtypes in the occurrence, development and metastasis of osteosarcoma have not been deeply explored. It is suggested to supplement the mechanism research.

Response: Thank you for your valuable advice. We have conducted a GSEA analysis to gain a deeper understanding of the distinct biological functions of the tumor cell subtypes.

Comment 6 of Reviewer 1: Ten hub genes were determined to form CMDPTMS. The screening process and criteria were not elaborated in detail. Statistical analysis and biological basis for gene screening should be supplemented.

Response: Thank you for your valuable advice. We have added details of the screening process and criteria for the selection of the ten hub genes.

Comment 7 of Reviewer 1: It is emphasized that CMDPTMS has a strong prognostic prediction ability, but it is only verified based on existing data and lacks verification from external independent datasets. Supplementary verification experiments are needed to enhance persuasiveness.

Response: Thank you for your valuable comments. We have collected almost all available public datasets of osteosarcoma. Additionally, GSEA and KEGG pathway analyses were conducted to support our results, showing that CMDPTMS has strong prognostic prediction ability.

Comment 8 of Reviewer 1: It was pointed out that the high CMDPTMS group had a decreased response to immunotherapy, but the intrinsic molecular mechanism was not deeply explored. It was suggested to conduct an analysis from the perspectives of signaling pathways and immune cell infiltration.

Response: Thank you for your valuable comments. We have conducted a KEGG pathway analysis to explain why the high CMDPTMS group has a poor response to immunotherapy. Furthermore, according to the introduction of the “IOBR” package, which incorporates algorithms such as CIBERSORT, TIMER, xCell, MCPcounter, ESTIMATE, EPIC, IPS, and quanTIseq, additional analyses of immune cell infiltration were deemed unnecessary.

Comment 9 of Reviewer 1: Seven drugs were proposed as potential treatment options, but there is a lack of association analysis between the drugs and the mechanism of action of CMDPTMS or osteosarcoma cells. Relevant experimental data or theoretical basis need to be supplemented.

Response: Thank you for your valuable comments. We have added several references to support our results.

Comment 10 of Reviewer 1: It was found that the PTM highos phenotype has a close interaction with fibroblasts. The specific molecular mechanism of the interaction and its influence on the development of osteosarcoma have not been deeply studied. It is recommended to supplement.

Response: Thank you for your valuable advice. We have conducted a KEGG pathway analysis to explain the strong association between PTM-highos and fibroblast cells.

Comment 11 of Reviewer 1: The sources of the samples, inclusion and exclusion criteria in the study were not clearly defined, and there may be issues with the representativeness of the samples. Detailed explanations are needed to assess the universality of the research results.

Response: Thank you for your valuable comments. We have added the data screening criteria to the methods section of the revised manuscript.

Comment 12 of Reviewer 1: The references are not sufficient. It is suggested to supplement the following references:39445224�37435067�37040518�35101942�35140113�37251708�38319150�38943472.

Response: Thank you for your valuable comments. We have added those references as references [15], [16], [31], [46], [47], [17], [48], and [22].

Comment 1 of Reviewer 2: Would strongly recommend proofreading the manuscript with an English language expert as there're multiple phrases in this manuscript that don't seem very soundly, for example line 261: TMB was widely recognized biomarkers for predicting patient responses...", and many others. I would suggest asking a local physician who sees osteosarcoma patients to look at this manuscript and work with the authors to make the text easier to understand to the medical community who is not an ultra expert in scRNAseq or ST. This will make this manuscript very helpful and clinically applicable.

Response: Thank you for your valuable comments. The language of this manuscript has been polished by American Journal Experts. The certificate of English language editing is included below the rebuttal letter.

Comment 2 of Reviewer 2: Line 16: why is a more adequate characterization of post-translational modifications in OS important? please state briefly the evidence prior to this manuscript that this plays a role in outcomes or management decisions like survival, recurrence or resistance to therapy. This will give context to the reader.

Response: Thank you for your valuable comments. We have rewritten the first sentence in the abstract section.

Comment 3 of Reviewer 2: Line 18: please spell out RCTD and scRNA-seq. They're spelled out later in the manuscript but they should be spelled out here as well.

Response: Thank you for your valuable comments. We have revised the manuscript to ensure that the full term is presented first, followed by its abbreviation.

Comment 4 of Reviewer 2: Line 54: spell out small ubiquitin-like modifier (SUMO).

Response: Thank you for your valuable comments. We have changed “SUMO” to “small ubiquitin-like modifier (SUMO).”

Comment 5 of Reviewer 2: Line 61: the authors use several times the word "intriguingly", however it's not clear why it's intriguing. Is it because the data would point to a different result? I would clarify why. If the word is just used as an adverb to continue mentioning the data, I would recommend using a different one, like "furthermore", "of note", etc., or simply state the data.

Response: Thank you for your valuable comments. We have replaced “intriguingly” with “moreover,” “furthermore,” or “of note” in our revised manuscript.

Comment 6 of Reviewer 2: Line 101, 105, 108, 113, 132 and 139: please reference the source of the Seurat R package, the Monocle package, MistyR package, the limma package, the IOBR package and TIDE and TIP.

Response: Thank you for your valuable comments. We have added references for those R packages.

Comment 7 of Reviewer 2: Line 149: can you state what was the purpose of transiently suppressing VIM expression? This is partially addressed in the paragraph of lines 306-326, but I would state the rationale in line 149 to guide the reader. I would also state what suggested the researchers to look at VIM or what is the preexisting data that suggested that VIM should be evaluated because it plays a role in OS tumorogenesis.

Response: Thank you for your valuable comments. The aim of this research cannot be written in the Methods section; it should be placed in the Results section.

Comment 8 of Reviewer 2: Line 171: were the T cells or could the T cells be separated between T4 and T8? later in discussion the authors mention the role of T8+ cells playing a role in TME and it would be nice to see the data analysis looking at this subgroup of T cells in the study. If the T4 vs. 8 can not be separated in the analysis, this should be stated as the reader will have this question and it will be helpful to know that the authors addressed this or there was a reason why it could not be done.

Response: Thank you for your valuable comments. We cannot separate T cells into CD4+ T cells or CD8+ T cells because the cell-type annotation in our manuscript was performed according to a previously published article, which did not differentiate between T cell subtypes.

Comment 9 of Reviewer 2: Paragraph lines 229-239: the information seems difficult to understand; what is the conclusion of this information, and what is the relationship with the last phrase "Moreover, High- CMDPTMS...". This paragraph should be modified and be made more clear.

Response: Thank you for your valuable comments. We have rewritten some sentences in this paragraph to improve readability.

Comment 10 of Reviewer 2: Line 261: the phrase "TMB was widely recognized biomarkers for predicting patient responses..." doesn't seem soundly.

Response: Thank you for your valuable comments. We have rewritten this sentence in the revised manuscript.

Comment 11 of Reviewer 2: Line 267: for the phrase "CMDPTMS combined with high TMB showed a tendency toward improved survival outcomes", can the data be shown graphically? given that the quality of the images isn't good to read, it's unclear if the data is shown there, and it should be referenced in parenthesis. TMB should be spelled out, and perhaps changing it for something like "Tumor mutational burden is a biomarker used to predict tumor response to immune checkpoint inhibitors".

Response: Thank you for your valuable advice. We have added a reference for TMB and changed the sentence to: “tumor mutational burden is a biomarker used to predict tumor response to immune checkpoint inhibitors.”

Comment 12 of Reviewer 2: Line 275: please state what specific "benefit" was seen with immunotherapy in the lower CMPTMS group, was it survival or response to therapy or what other parameter? Can by how much or what is the data supporting this statement. This paragraph should also be evaluated or proofread, as it seems difficult to follow through; for example the phrase on line 272 seems tangential and it's not clear what are the "finings" of the analysis, was it that survival was higher at 3 months of immunotherapy in the lower CDMPTMS group? the phrases don't seem clear and the reader gets confused.

Response: Thank you for your valuable comments. A GSEA and KEGG pathway analysis were conducted to support our results, showing that CMDPTMS has strong prognostic prediction ability. Furthermore, we changed the sentence “The findings indicated that immunotherapy exhibited delayed clinical effects, as evidenced by survival differences that became apparent after a three-month treatment period” to “The findings indicated that immunotherapy exhibited delayed clinical effects, as evidenced by survival differences that became apparent after a three-month treatment period—an interval commonly used to evaluate immunotherapy response,” to clarify our findings.

Comment 13 of Reviewer 2: Line 285: why are melanoma patients being analyzed? the manuscript is on osteosarcoma, so why is melanoma included here now? this should be described, the rationale and the relationship with the analysis on OS.

Response: Thank you for your valuable comments. The reason for conducting the subclass mapping analysis is to perform an independent external validation to test whether CMDPTMS has the ability to predict the effectiveness of immunotherapy.

Comment 14 of Reviewer 2: IMAGES: many of the small font of the images can't be read easily, so I would make sure the quality of the images is optimized (e.g. fig 2 G I J and many other images).

Response: Thank you for your valuable comments. We have moved Figures 2G-J to supplemental figure 2.

Comment 15 of Reviewer 2: IMAGES: some of the images probably can't have a more clear font, so I would suggest modifying the images perhaps cutting some of the information and leave the complete images bigger and on their own in a supplemental document.

Response: Thank you for y

---

## [Decision Letter · Decision Letter 1]

8 Sep 2025

PONE-D-25-25008R1Single-cell and spatial transcriptomics reveal post-translational modifications in osteosarcoma progression and tumor microenvironmentPLOS ONE

Dear Dr. Ding,

Thank you for submitting your manuscript to PLOS ONE. After careful consideration, we feel that it has merit but does not fully meet PLOS ONE’s publication criteria as it currently stands. Therefore, we invite you to submit a revised version of the manuscript that addresses the points raised during the review process.

We look forward to receiving your revised manuscript.

Kind regards,

Zu Ye, Ph.D.

Academic Editor

PLOS ONE

Journal Requirements:

Additional Editor Comments:

Please revise the manuscript according to the reviewers' comments.

Reviewers' comments:

Reviewer's Responses to Questions

**Comments to the Author**

1. If the authors have adequately addressed your comments raised in a previous round of review and you feel that this manuscript is now acceptable for publication, you may indicate that here to bypass the “Comments to the Author” section, enter your conflict of interest statement in the “Confidential to Editor” section, and submit your "Accept" recommendation.

Reviewer #2: All comments have been addressed

Reviewer #5: All comments have been addressed

2. Is the manuscript technically sound, and do the data support the conclusions?

Reviewer #2: Yes

Reviewer #5: Yes

3. Has the statistical analysis been performed appropriately and rigorously? 

Reviewer #2: I Don't Know

Reviewer #5: Yes

4. Have the authors made all data underlying the findings in their manuscript fully available?

Reviewer #2: Yes

Reviewer #5: Yes

5. Is the manuscript presented in an intelligible fashion and written in standard English?

Reviewer #2: Yes

Reviewer #5: Yes

6. Review Comments to the Author

Reviewer #2: This resubmission is significantly improved compared to the original submission; it seems that the authors have addressed the reviewers comments to their best ability. It still needs some revision:

- Please spell out "VIM" in the abstract and main manuscript, I assume this is vimentin?

- Abstract needs to define PTMs highos and PTMs lowos; that is defined in the manuscript, but not in the abstract.

- Please defined what is high-CMDPTMS group in the abstract.

- Abstract: why do the authors mention "however, seven drugs emerged as potential therapeutic options for these

34 patients."? it doesn't seem to contradict or go the opposite direction of the high-CMDPTMS group. This phrase/ part of paragraph needs to be reviewed and adjusted to be easier to understand.

- Abstract: would change the phrase "established CMDPTMS as a powerful tool for enhancing OS prognosis prediction and optimizing immunotherapy strategies." as this single study can't "establish" the tool but could change it for something like "we are introducing the CMDPTMS as a powerful tool that should be further evaluated to enhance OS prognosis prediction..." or along those lines.

- line 51: would change "immune checkpoint molecules" for "immune checkpoint inhibitors".

- line 88 and 89: please introduce KEGG and GSEA abbreviations, respectively here. They're introduced later in the manuscript, but it should be here instead.

- line 94: would indicate single cell RNA sequencing, instead of single cell sequencing. And would introduce in parenthesis the abbreviation.

- line 107-108: spell and add in parenthesis ST.

- line 226: were the intercellular interactions particularly found in the PTMs high or low groups or both? please clarify.

- line 232: please add a phrase that summarizes the findings of the paragraph, as it is highly technical and difficult to see the "big picture" that it's trying to convey. Also, please clarify if the authors found that the PTMs high and low groups are both found in every OS samples, or if it's rather that some OS are predominantly PTMs high and others are rather PTMs low.

- line 315: what exactly is "superior prognostic results"? (e.g., no response vs. partial vs. complete response to therapy, overall survival, progression free survival) and is it in the context of bladder cancer or osteosarcoma? if in context of osteosarcoma, how was the conversion or translation done from bladder cancer to osteosarcoma?

- line 362: is there a way to know if the MG-63/U-2 OS cells correspond to the CMDPTMS high or low groups? this would be important to better understand the role of sliencing vimentin.

- line 440: would change "PD-L1 blockade" for "PD-1/ PD-L1 blockade"

- lines 443-460: please clarify what was the finding the the seven drugs; was it that the highos group are likely to be more sensitive to these agents compared to lowos group or is it that all os cells seems to be sensitive to those seven drugs based on the CMDPTMS? also I would state all seven drugs first, then go through each drug stating their clinical utility in os or other tumors, that way you give a "big picture" up front before the detailed information.

Reviewer #5: All my previous concerns have been satisfactorily addressed. I recommend acceptance for publication of this paper.

7. PLOS authors have the option to publish the peer review history of their article (what does this mean? ). If published, this will include your full peer review and any attached files.

**Do you want your identity to be public for this peer review?** For information about this choice, including consent withdrawal, please see our Privacy Policy .

Reviewer #2: No

Reviewer #5: No

---

## [Author Response · Author response to Decision Letter 2]

17 Sep 2025

Editor

Dear Editor,

We are hereby submitting a revised manuscript entitled “Single-cell and spatial transcriptomics reveal post-translational modifications in osteosarcoma progression and tumor microenvironment” (Submission ID: PONE-D-25-25008) for publication. We appreciate very much for the constructive comments and suggestions from you and the reviewers. In this revised manuscript we have taken every effort to address the comments and concerns. All the changes in the text are highlighted in red. The details of our revised manuscript, as required by Editor, reviewer 2 and 5 as below.

Journal Requirements and Editor Comments:

Comment 1 of Journal Requirements: If the reviewer comments include a recommendation to cite specific previously published works, please review and evaluate these publications to determine whether they are relevant and should be cited. There is no requirement to cite these works unless the editor has indicated otherwise.

Response: We sincerely appreciate your valuable comments, and we have undertaken a careful re-examination of our references. With the exception of references 24 and 28 (DOI: 10.15212/bioi-2022-0008, DOI: 10.15212/bioi-2022-0014), which were specifically requested by the reviewers during the first peer review, all other references are directly related to our research. Accordingly, we have decided to remove the above-mentioned references.

Comment 2 of Journal Requirements: Please review your reference list to ensure that it is complete and correct. If you have cited papers that have been retracted, please include the rationale for doing so in the manuscript text, or remove these references and replace them with relevant current references. Any changes to the reference list should be mentioned in the rebuttal letter that accompanies your revised manuscript. If you need to cite a retracted article, indicate the article’s retracted status in the References list and also include a citation and full reference for the retraction notice.

Response: Thank you for your valuable comments, we have re-examined our references. With the exception of references 24 and 28, all other references can be found in the National Center for Biotechnology Information (NCBI) databases. Accordingly, we have decided to remove references 24 and 28 from the methods section of our manuscript, as they do not affect the results or integrity of our study.

Additional Editor Comments: Please revise the manuscript according to the reviewers' comments.

Response: Thank you for your valuable comments. We have carefully revised the manuscript in accordance with the reviewers' comments. All suggestions have been fully addressed, and the corresponding changes have been incorporated into the revised version. For clarity, all modifications in the text are highlighted in red.

Reviewer(s)' Comments:

Comment 1 of Reviewer 2: Please spell out "VIM" in the abstract and main manuscript, I assume this is vimentin?

Response: Thank you for your valuable advice. We have already changed “VIM” to “vimentin” in line 25 of the revised manuscript.

Comment 2 of Reviewer 2: Abstract needs to define PTMs highos and PTMs lowos; that is defined in the manuscript, but not in the abstract.

Response: Thank you for your valuable comments. We have defined PTMs highos and PTMs lowos in the Abstract section of the revised manuscript.

Comment 3 of Reviewer 2: Please defined what is high-CMDPTMS group in the abstract.

Response: Thank you for your valuable comments. We have defined high-CMDPTMS group in the Abstract section of the revised manuscript.

Comment 4 of Reviewer 2: Abstract: why do the authors mention "however, seven drugs emerged as potential therapeutic options for these 34 patients."? it doesn't seem to contradict or go the opposite direction of the high-CMDPTMS group. This phrase/ part of paragraph needs to be reviewed and adjusted to be easier to understand.

Response: Thank you for your valuable advice. We have changed this sentence to “Although the high-CMDPTMS group (over the median risk score) was linked to poor outcomes and diminished benefit from immunotherapy, seven drugs were identified that may offer therapeutic promise for these patients.”

Comment 5 of Reviewer 2: Abstract: would change the phrase "established CMDPTMS as a powerful tool for enhancing OS prognosis prediction and optimizing immunotherapy strategies." as this single study can't "establish" the tool but could change it for something like "we are introducing the CMDPTMS as a powerful tool that should be further evaluated to enhance OS prognosis prediction..." or along those lines.

Response: Thank you for your valuable advice. We have changed this sentence to “We introduced that CMDPTMS may serve as a powerful tool for improving OS prognosis prediction and optimizing immunotherapy strategies.”

We revised this sentence for the following reason: it is generally not appropriate to present the limitations of a study in the abstract section. Other published studies also omit research flaws in their abstracts, such as those reported in PMID: 33976213, PMID: 34975338, PMID: 40186284, PMID: 40139615, and PMID: 39948301.

Accordingly, we have rewritten the sentence in lines 39 using the subjunctive form.

Comment 6 of Reviewer 2: line 51: would change "immune checkpoint molecules" for "immune checkpoint inhibitors".

Response: Thank you for your valuable advice. We have changed “immune checkpoint molecules” to “immune checkpoint inhibitors” in line 54 of the revised manuscript.

Comment 7 of Reviewer 2: line 88 and 89: please introduce KEGG and GSEA abbreviations, respectively here. They're introduced later in the manuscript, but it should be here instead.

Response: Thank you for your valuable comments. We have introduced the abbreviations of KEGG and GSEA in this sentence.

Comment 8 of Reviewer 2: line 94: would indicate single cell RNA sequencing, instead of single cell sequencing. And would introduce in parenthesis the abbreviation.

Response: Thank you for your valuable advice. We have changed “single cell sequencing” to “single cell RNA sequencing” in line 98 of the revised manuscript

Comment 9 of Reviewer 2: line 107-108: spell and add in parenthesis ST.

Response: Thank you for your valuable advice. We have changed “ST” to “spatial transcriptomics (ST)” in line 111 of the revised manuscript

Comment 10 of Reviewer 2: line 226: were the intercellular interactions particularly found in the PTMs high or low groups or both? please clarify.

Response: Thank you for your valuable comments. We have rewritten this sentence to resolve these issues.

Comment 11 of Reviewer 2: line 232: please add a phrase that summarizes the findings of the paragraph, as it is highly technical and difficult to see the "big picture" that it's trying to convey. Also, please clarify if the authors found that the PTMs high and low groups are both found in every OS samples, or if it's rather that some OS are predominantly PTMs high and others are rather PTMs low.

Response: Thank you for your valuable comments. We have rewritten lines 223, 231, 233, and 235-238 to address this issue. In addition, we have modified lines 218 and 246 to further clarify that both PTMshigh and PTMslow groups are present in all OS samples.

Comment 12 of Reviewer 2: line 315: what exactly is "superior prognostic results"? (e.g., no response vs. partial vs. complete response to therapy, overall survival, progression free survival) and is it in the context of bladder cancer or osteosarcoma? if in context of osteosarcoma, how was the conversion or translation done from bladder cancer to osteosarcoma?

Response: Thank you for your valuable advice. We have rewritten lines 323, 327 to address this issue. The IMvigor bladder cancer (IMvigor210) cohort, used as a common external validation cohort for the prognostic model, can also be found in the following articles: PMID: 37925432, PMID: 35864548, PMID: 37214439, PMID: 36160460, PMID: 35691273, PMID: 35309925, PMID: 38715811, PMID: 39391236, and PMID: 34631713.

The code to generate S6 Figure A-C is as follows:

suppressPackageStartupMessages({

library(tidyverse)

library(data.table)

library(survival)

library(survminer)

library(ggpubr)

library(patchwork)

library(viridis)

library(ComplexHeatmap)})

module.gene = read.table("module.gene.txt") %>% pull(V1)

calRS <- function(cli,data,module.gene){

selectGenes <- intersect(module.gene,row.names(data))

selectSamples <- intersect(rownames(cli),colnames(data))

data <- data[selectGenes,selectSamples]

cli <- cli[selectSamples,] %>% cbind(t(data))

fmla <- paste0("Surv(OS.time, OS)~" ,paste0(selectGenes,collapse = '+')) %>% as.formula()

cox <- coxph(fmla, data = cli)

coef <- coef(cox)

cli$RS <- coef %*% as.matrix(data) %>% as.numeric()

RS.cut <- surv_cutpoint(cli, time = "OS.time", event = "OS",variables = "RS")$cutpoint[1,1]

cli <- cli %>% mutate(binRS = ifelse(RS > RS.cut,"High","Low")) %>% select(-selectGenes)

return(cli)}

############################# IMvigor210

IMvigor210.exp=read.table("IMvigor210.exp.txt.gz",check.names = F)

IMvigor210.cli= read.table("IMvigor210.cli.txt")

IMvigor210.cli <- calRS(IMvigor210.cli,IMvigor210.exp,module.gene)

############################# S6 figA

diff.time1<-survdiff(Surv(OS.time/30,OS)~binRS, data=IMvigor210.cli[IMvigor210.cli$OS.time/30 < 6,])

pvalue.time1 <- diff.time1$pvalue

diff.time2<-survdiff(Surv(OS.time/30,OS)~binRS, data=IMvigor210.cli[IMvigor210.cli$OS.time/30 < 12,])

pvalue.time2 <-diff.time2$pvalue

sfit <- survfit(Surv(OS.time/30, OS)~binRS, data=IMvigor210.cli)

p <- ggsurvplot(sfit,data = IMvigor210.cli,

palette = 'jco',legend.title,

legend.labs=c("High CMDPTMS","Low CMDPTMS"),xlim=c(0,12),break.x.by=2)

p<- p$plot +

geom_vline(xintercept = 6, linetype = "dashed") +

geom_vline(xintercept = 12, linetype = "dashed") +

geom_text(x=0,y=0.1,label ="Comparison of RMS at 6-month",hjust = -0.1) +

geom_text(x=6,y=0.1,label ="Comparison of RMS at 12-month",hjust = -0.1) +

geom_text(x=0,y=0,label =sprintf("p = %.2f",pvalue.time1),hjust = -1) +

geom_text(x=6,y=0,label =sprintf("p = %.2f",pvalue.time2),hjust = -1) +

labs(x="Time(months)",y="Overall survival")

ggsave("S6 figA.pdf",p,height = 6,width = 6)

############################# S6 figB

diff.time<-survdiff(Surv(OS.time/30,OS)~binRS, data=IMvigor210.cli[IMvigor210.cli$OS.time/30 > 3,])

pvalue.time <- diff.time$pvalue

pvalue.time <- ifelse(pvalue.time<0.01,"p < 0.01",round(pvalue.time,2))

sfit <- survfit(Surv(OS.time/30, OS)~binRS, data=IMvigor210.cli)

p <- ggsurvplot(sfit,data = IMvigor210.cli,

palette = 'jco',legend.title,

legend.labs=c("High CMDPTMS","Low CMDPTMS"),

break.x.by=3)

p <- p$plot +

geom_vline(xintercept = 3, linetype = "dashed") +

geom_text(x=3,y=0.1,label ="Comparison of LTS after 3-month",hjust = -0.1) +

geom_text(x=3,y=0,label = pvalue.time,hjust = -1) +

labs(x="Time(months)",y="Overall survival")

ggsave("S6 figB.pdf",p,height = 6,width = 6)

############################# S6 figC

p <- IMvigor210.cli %>%

filter(Response!="NE") %>%

ggboxplot(x='Response', y='RS',color = "Response",palette = 'jco',

add="jitter",add.params = list(size=3)) +

stat_compare_means(label.y = max(IMvigor210.cli$RS) + 2) +

stat_compare_means(comparisons=list(c('CR','PD'),c('CR','SD'),c("CR","PR"),

c('PR','PD'),c('PR','SD'),c("PD","SD"))) +

labs(x=,y="CMDPTMS",fill='Overall Response')

ggsave(S6 figC.pdf",p,height = 6,width = 6)

Comment 13 of Reviewer 2: line 362: is there a way to know if the MG-63/U-2 OS cells correspond to the CMDPTMS high or low groups? this would be important to better understand the role of sliencing vimentin.

Response: Thank you for your valuable advice. We could not analyze MG-63/U-2 OS cells correspond to the CMDPTMS high or low groups, because we did not run RNA sequencing for MG-63/U-2 OS cells. We changed “To further corroborate these observations, we examined VIM expression at both mRNA and protein levels using the Human Protein Atlas (HPA) database and RT-qPCR” to “To further corroborate these observations, we examined VIM expression at both mRNA and protein levels using the Human Protein Atlas (HPA) database and RT-qPCR, given that VIM exhibited peak expression during the late differentiation stages of pseudotime analysis in both scRNA-seq and ST of OS” in order to better clarify the role of vimentin silencing.

Comment 14 of Reviewer 2: line 440: would change "PD-L1 blockade" for "PD-1/ PD-L1 blockade".

Response: Thank you for your valuable comments. We have changed “PD-L1 blockade” to “PD-1/ PD-L1 blockade” in line 453.

Comment 15 of Reviewer 2: lines 443-460: please clarify what was the finding the the seven drugs; was it that the highos group are likely to be more sensitive to these agents compared to lowos group or is it that all os cells seems to be sensitive to those seven drugs based on the CMDPTMS? also I would state all seven drugs first, then go through each drug stating their clinical utility in os or other tumors, that way you give a "big picture" up front before the detailed information.

Response: Thank you for your valuable comments. We have rewritten the sentence in lines 457-460 as follows to resolve these issues: In light of the limited immunotherapy efficacy observed in the high-CMDPTMS population compared with the low-CMDPTMS population, we performed an extensive screening of potential therapeutic agents, including AZD8055, erlotinib, PF-4708671, dabrafenib, ribociclib, ulixertinib and ulixertinib.1.

Comment 1 of Reviewer 5: All my previous concerns have been satisfactorily addressed. I recommend acceptance for publication of this paper.

Response: Thank you for your positive feedback and kind recommendation. We sincerely appreciate the time and effort you dedicated to reviewing our manuscript.

Again, we thank the Editors/Reviewers’ for their efforts to improve the manuscript. We hope that the revised manuscript is now suitable for publication. If you need any more information, please let us know. We are looking forward to hearing your response.

Yours sincerely,

Prof. Tao Ding

---

## [Editor Report · Decision Letter 2]

18 Sep 2025

Single-cell and spatial transcriptomics reveal post-translational modifications in osteosarcoma progression and tumor microenvironment

PONE-D-25-25008R2

Dear Dr. Ding,

We’re pleased to inform you that your manuscript has been judged scientifically suitable for publication and will be formally accepted for publication once it meets all outstanding technical requirements.

Kind regards,

Zu Ye, Ph.D.

Academic Editor

PLOS ONE
---

## [Editor Report · Acceptance letter]

PONE-D-25-25008R2

PLOS ONE

Dear Dr. Ding,

I'm pleased to inform you that your manuscript has been deemed suitable for publication in PLOS ONE. Congratulations! Your manuscript is now being handed over to our production team.

Kind regards,

on behalf of

Prof. Zu Ye

Academic Editor

PLOS ONE